# Thermalization and criticality on an analogue–digital quantum simulator

Understanding how interacting particles approach thermal equilibrium is a major challenge of quantum simulators[1,2]. Unlocking the full potential of such systems towards this goal requires flexible initial state preparation, precise time evolution and extensive probes for final state characterization. Here we present a quantum simulator comprising 69 superconducting qubits that supports both universal quantum gates and high-fidelity analogue evolution, with performance beyond the reach of classical simulation in cross-entropy benchmarking experiments. This hybrid platform features more versatile measurement capabilities compared with analogue-only simulators, which we leverage here to reveal a coarsening-induced breakdown of Kibble–Zurek scaling predictions[3] in the *XY* model, as well as signatures of the classical Kosterlitz–Thouless phase transition[4]. Moreover, the digital gates enable precise energy control, allowing us to study the effects of the eigenstate thermalization hypothesis[5–7] in targeted parts of the eigenspectrum. We also demonstrate digital preparation of pairwise-entangled dimer states, and image the transport of energy and vorticity during subsequent thermalization in analogue evolution. These results establish the efficacy of superconducting analogue–digital quantum processors for preparing states across many-body spectra and unveiling their thermalization dynamics.

The advent of quantum simulators in various platforms[8–14] has opened a powerful experimental avenue towards answering the theoretical question of thermalization[5,6], which seeks to reconcile the unitarity of quantum evolution with the emergence of statistical mechanics in constituent subsystems. A particularly interesting setting is that in which a quantum system is swept through a critical point[15–18], as varying the sweep rate can allow for accessing markedly different paths through phase space and correspondingly distinct coarsening behaviour. Such effects have been theoretically predicted to cause deviations[19–22] from the celebrated Kibble–Zurek (KZ) mechanism, which states that the correlation length $\xi$ of the final state follows a universal power-law scaling with the ramp time $t_r$ (refs. 3,23–25).

Whereas tremendous technical advancements in quantum simulators have enabled the observation of a wealth of thermalization-related phenomena[26–35], the analogue nature of these systems has also imposed constraints on the experimental versatility. Studying thermalization dynamics necessitates state characterization beyond density–density correlations and preparation of initial states across the entire eigenspectrum, both of which are difficult without universal quantum control[36]. Although digital quantum processors are in principle suitable for such tasks, implementing Hamiltonian evolution requires a high number of digital gates, making large-scale Hamiltonian simulation infeasible under current gate errors.

In this work, we present a hybrid analogue–digital[37,38] quantum simulator comprising 69 superconducting transmon qubits connected by tunable couplers in a two-dimensional (2D) lattice (Fig. 1a). The quantum simulator supports universal entangling gates with pairwise interaction between qubits, and high-fidelity analogue simulation of a $U(1)$ symmetric spin Hamiltonian when all couplers are activated

at once. The low analogue evolution error, which was previously difficult to achieve with transmon qubits due to correlated cross-talk effects, is enabled by a new scalable calibration scheme (Fig. 1b). Using cross-entropy benchmarking (XEB)[39], we demonstrate analogue performance that exceeds the simulation capacity of known classical algorithms at the full system size.

Leveraging these capabilities, we prepare and characterize states of a 2D *XY* magnet with broadly tunable energy density, allowing us to study the interplay between quantum and classical critical behaviour in the rich phase diagram of our system. Specifically, we observe finite-size signatures of the Kosterlitz–Thouless topological phase transition—including the emergence of algebraically decaying correlations with exponent near the expected universal value of $\frac{1}{4}$—and demonstrate a breakdown of the KZ mechanism. Our study takes advantage of extensive measurement capabilities to characterize, for example, entanglement entropy for subsystems up to 12 qubits, multi-qubit vortex correlators and energy fluctuations. We also leverage our hybrid analogue–digital scheme (Fig. 1a) to prepare entangled initial states, allowing us to spatially tailor the energy density and vorticity, and investigate the subsequent thermalization dynamics and energy transport.

Operating coupled transmons as a high-fidelity analogue quantum simulator requires precise knowledge of the many-body spin Hamiltonian $H_s$, which depends on the 'bare' frequencies, $\{\omega_{qi}\}$ and $\{\omega_{cj}\}$, of qubits $qi$ and couplers $cj$. However, experimental calibration is only capable of resolving 'dressed' frequencies that—unlike the bare frequencies—change from local (isolated) calibrations to full-scale experiments due to hybridization with neighbouring qubits and couplers. Given this difficulty, past experimental studies[30,31] either suffered from large

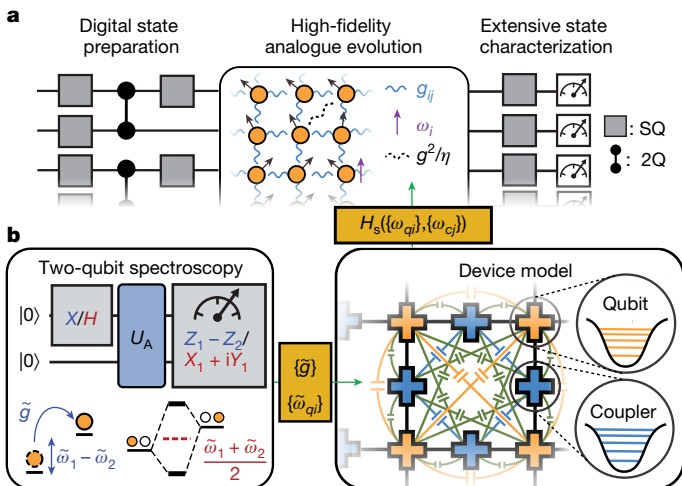

**Fig. 1 | Analogue–digital simulation with high-precision calibration. a**, Our platform combines analogue evolution with digital gates for extensive state preparation and characterization. **b**, Schematic of new scalable analogue calibration scheme. Swap (blue) and single-photon (red) spectroscopy is used to extract dressed coupling rates ($\{\widetilde{g}\}$) and single frequencies ($\{\widetilde{\omega}_{qi}\}$) of two-qubit analogue evolution ($U_A$), which are converted to bare qubit and coupler frequencies ($\{\omega_{qi}\}, \{\omega_{cj}\}$) through detailed device modelling. The bare frequencies allow for establishing the device Hamiltonian of the full system, which is finally projected to a spin Hamiltonian, $H_s$.

errors or resorted to multi-parameter optimization protocols that are difficult to scale up. Sophisticated Hamiltonian learning techniques[40,41] can circumvent these issues, but still have potential vulnerabilities to Hamiltonian ramps, noise and errors in state preparation and measurement (SPAM).

In this work, we present a scalable calibration protocol that achieves low error by explicitly calibrating the bare frequencies. As illustrated in Fig. 1b, the protocol begins with two-qubit calibration measurements (single-photon and swap spectroscopy, which is robust to ramps and SPAM errors) to determine the effective coupling $\widetilde{g}$ and dressed qubit frequencies $\{\widetilde{\omega}_{qi}\}$ of every qubit pair. Next, we use extensive modelling of the underlying device physics to convert the dressed quantities to the bare frequencies $\{\omega_{qi}\}, \{\omega_{cj}\}$. Finally, a projection technique is applied

to approximate our high-dimension device Hamiltonian, $H_d(\{\omega_{qi}\}, \{\omega_{cj}\})$, into a spin Hamiltonian, $H_s$:

$$H_s = \sum_i \omega_i n_i + \sum_{\langle i,j \rangle} g_{ij}(X_i X_j + Y_i Y_j)/2 + \mathcal{O}(g^2/\eta) \tag{1}$$

where $\omega_i$ and $|g_{ij}| \approx g$ are tunable on-site potentials and nearest-neighbour couplings, respectively. The latter is notably smaller than the qubit anharmonicity $\eta \gg g$. This restricts the photon occupation numbers to $n_i = 0, 1$ and $X_i, Y_i$ are Pauli operators acting in this subspace. The Hamiltonian in equation (1) is in the universality class of an *XY* model with on-site *z*-fields. A natural consequence of the hybridization in our system is that $H_s$ contains not only nearest-neighbour hopping, but also density–density interactions and next-nearest-neighbour terms, which scale as order $\mathcal{O}(g^2/\eta)$ and are typically five to ten times smaller than *g* (see further details in Methods).

A computationally challenging problem and useful benchmark for the quantum simulator is the thermalization dynamics of an initial *Z*-basis product state at half-filling, which has high temperature with respect to $H_s$ and hosts many quasiparticles (Fig. 2a). When subject to the (photon number conserving) time evolution operator $e^{-iH_s t/\hbar}$ where $\hbar$ is the reduced Planck constant (set to 1 hereafter), interactions between quasiparticles are expected to drive the system into a chaotic state. To explore these dynamics, we perform a rapid (6 ns) ramp of the couplings $g_{ij}/2\pi$ from 0 to 10 MHz. Quantum chaotic behaviour is then diagnosed by means of *Z*-basis measurements at different times, yielding a set of probability distributions $p_{meas}(x, t)$ where $\{x\}$ represents the set of *D* 'bitstrings' with the same number of photons as the initial state. Figure 2b shows the distribution $\Pr(p)$ of $p_{meas}(x, t)$ for reduced system sizes up to $N_q = 25$ at $t = 5.5/g$. In each case, $\Pr(p)$ shows a clear exponential decay known as the Porter–Thomas distribution, signalling thermalization to a quantum chaotic state[39,42]. By contrast, past studies have found substantial deviations from the Porter–Thomas distribution in other models of analogue dynamics[43,44].

Characterizing the thermalization dynamics through the second moment of the bitstring distribution, also called the self-XEB[39], $D\sum_x p_{meas}^2(x, t) - 1$, we observe its fast convergence to the Porter–Thomas (PT) value of 1 within $t_{PT} \approx 60$ ns (roughly $4/g$) for all system sizes (Fig. 2b inset, see Supplementary Information for similar saturation rate of entanglement entropy). The observed fast scrambling dynamics are due to the simultaneously activated couplers,

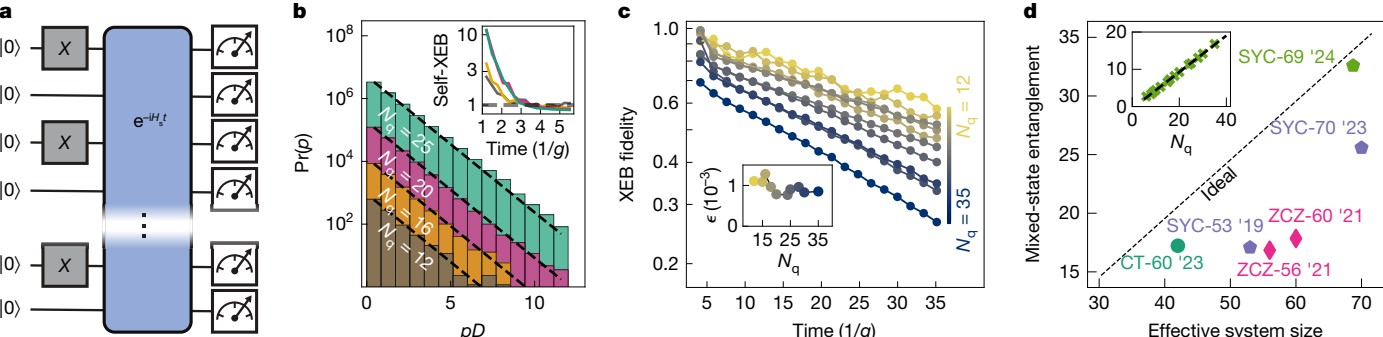

**Fig. 2 | Fast thermalization dynamics and beyond-classical capabilities in the high-temperature regime. a**, Schematic representation of the experiment: $N_q$ qubits are initialized in a half-filling state, evolved under a Hamiltonian $H_s$ over time *t* with four instances of disorder in $\{\omega_i\}$, and finally measured in the *Z*-basis. **b**, Distribution $\Pr(p)$ of bitstring probabilities *p* from experiment (coloured bars) at $t = 96$ ns $\approx 5.5/g$ and ideal Porter–Thomas distribution $\Pr(p) = De^{-pD}$ (dashed lines). Inset, convergence of the self-XEB with time. **c**, Time-dependent XEB fidelity for system sizes up to $N_q = 35$. Inset, system size dependence of $\epsilon$ (error per qubit per evolution time of $1/g$) from exponential fits. **d**, Mixed-state entanglement proxy, $\mathcal{E}_P$, obtained in this and previous

studies, plotted against the effective system size $N_q^{eff}$ (with respect to entanglement of a fully chaotic state; Supplementary Information) of the respective platforms. Blue pentagons, Sycamore processor in the digital regime[45,48]; diamonds, Zuchongzhi processor[46,47]; circle, neutral atom analogue simulator[44]; green pentagon, present experiment. $N_q^{eff}$ is equal to the actual $N_q$ in the digital experiments, whereas analogue platforms are subject to $U(1)$ conservation (this work) or constraints from Rydberg blockade[44]. Inset, $\mathcal{E}_P$ as a function of $N_q$ computed from experimental data, including the linear fit used for extrapolation to 69 qubits.

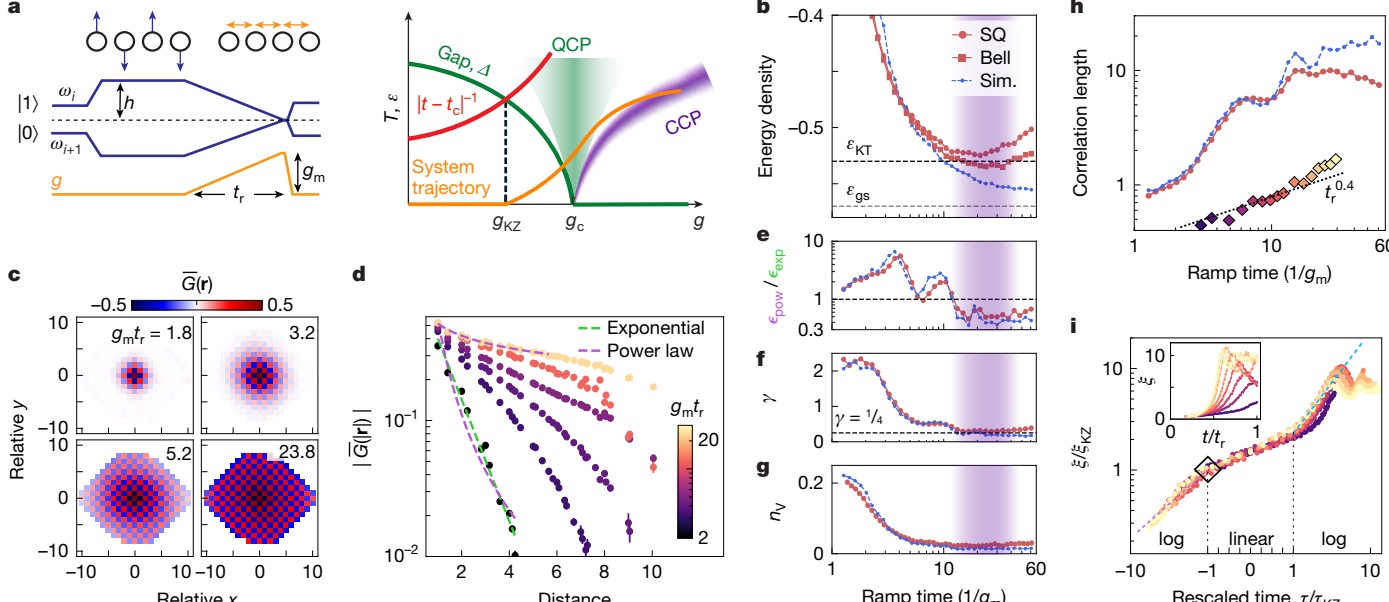

**Fig. 3 | Critical coarsening in the *XY* model. a**, Left, experimental schematic of qubit frequencies (blue) and coupling (yellow). Right, phase diagram. Dynamics become diabatic (dashed black) with increased temperature (*T*) when inverse remaining time (red) exceeds gap (green; $\Delta \propto |g - g_c|^{\nu z}$). QCP and CCP denote the quantum and classical critical phases, respectively. **b**, The final energy density approaches the ground state value ($\varepsilon_{gs}$, grey) and Kosterlitz–Thouless transition value ($\varepsilon_{KT}$, black) as $t_r$ is increased. Red circles and squares indicate single-qubit (SQ) and Bell basis measurements, respectively. Blue, MPS simulation. Purple shading indicates where classical critical behaviour is expected. **c**, Average correlation, $\overline{G}(\mathbf{r})$ (found from averaging $(\langle X_i X_j + Y_i Y_j \rangle - \langle X_i \rangle \langle X_j \rangle - \langle Y_i \rangle \langle Y_j \rangle)/2$ over all pairs *i*,*j* separated by **r**) measured at various $t_r$. **d**, Decay of radially averaged correlations. Green and purple curves show examples of exponential and power-law fits, respectively, performed up to a distance of 6 to avoid finite-size effects at longer distances. Error bars represent one standard deviation estimated from bootstrapping ($N_{reps} = 5 \times 10^4$ repetitions). **e**, Ratio between

r.m.s. errors from power-law and exponential fits ($\epsilon_{pow}/\epsilon_{exp}$) decreases for $g_m t_r > 15$. **f**, Power-law exponent, $\gamma$, approaches expected value at Kosterlitz–Thouless transition (1/4; black line). **g**, Vortex density proxy, $n_v$, decreases to minimum of $2 \times 10^{-2}$. **h**, Correlation length increases with $t_r$. Both simulation results (blue) and experimental data (red) show substantially more superlinear growth than KZ predictions (dashed black). Diamonds, correlation lengths extracted at expected freezing point (**i**). **i**, Correlation length during the ramp, shown with and without rescaling (main and inset, respectively) and two-sided logarithmic axes for $|\tau| > \tau_{KZ}$. $\xi_{KZ}$ is found from fitting $\xi(\tau = \tau_{KZ}) = \xi_0 (\tau_{KZ}/t_r)^{-\beta}$ with $\beta = 0.9(1)$ (difference from $\beta = \nu = 0.67$ expected to be due to finite-size effects). Dashed coloured lines show the theoretically expected scaling, $f(x) = |x|^{-\nu}$ with $\nu = 0.67$ for $x < -1$ (purple) and a heuristic $f(x) = x^\eta$ with $\eta = 1$ for $x > 1$ (teal). The increase in $\xi$ beyond the freezing point (diamond) causes deviation from KZ predictions.

and−compared to an equivalent digital circuit−allow for less shift towards the decohered distribution, $\Pr(p) = D^{-1}$ with self-XEB = 0.

To also characterize the coherent errors from imperfect calibration of $H_s$, we consider the linear XEB fidelity, $F(t) = D \sum_x p_{meas}(x, t) p_{sim}(x, t) - 1$, where $p_{sim}$ are exactly simulated probabilities[39]. The results, shown in Fig. 2c, show exponential decay after times roughly $t_{PT}$, where *F* accurately describes the state fidelity (Supplementary Information). Fitting the decay, we obtain an error rate of $\epsilon = 0.10 \pm 0.02\%$ per qubit per evolution time of $1/g$ (one cycle). $\epsilon$ is nearly independent of system size up to the largest exactly simulated system, $N_q = 35$ (inset of Fig. 2c). This indicates the scalability of our calibration protocol and allows extrapolation to the full system size of $N_q = 69$. Approximate matrix product state (MPS) simulations with bond dimension up to $\chi = 1,024$ were found to be ineffective beyond exactly simulatable system sizes, due to the fast entanglement growth and 2D geometry of our system (Supplementary Information).

The combination of the observed fast dynamics and high fidelity enables quantum simulation of computationally complex states. A representative metric of this capability is the mixed-state entanglement proxy, $\mathcal{E}_P = S_{ent}^{Rényi-1/2} + \log_2 F$, which lower bounds the mixed-state entanglement by accounting for the effects of infidelity on the pure-state Rényi-1/2 entropy[44]. Figure 2d compares the estimated $\mathcal{E}_P$ of our work and other recent state-of-the-art experiments[44–48], in which the proximity to the diagonal (ideal) line measures fidelity, indicating that our platform offers new possibilities for high-accuracy study of highly entangled states. In particular, we estimate that simulations with the level of our experimental fidelity requires more

than $10^6$ years on the Frontier supercomputer (Supplementary Information).

Having explored the thermalization dynamics in the high-temperature regime, we next turn to the rest of the rich phase diagram in the *XY* model (equation (1)), which is expected to show both a quantum phase transition in the ground state and a classical Kosterlitz–Thouless phase transition at finite temperature[4]. To prepare low-energy states of an antiferromagnetic *XY* magnet, we apply a staggered *z*-field of magnitude $h/(2\pi) = 30$ MHz and initialize the qubits in the *Z*-basis Neel state, maximizing the energy with respect to the first term in equation (1). We then ramp down the staggered field while simultaneously turning on ferromagnetic couplings of magnitude $g_m/(2\pi) = 20$ MHz over a duration $t_r$ (Fig. 3a). Under such a protocol[49], the system evolution is equivalent to that of an antiferromagnetic *XY* model with staggered field, initialized in the ground state. This ramp crosses a quantum phase transition between a paramagnetic phase with unbroken *U*(1) symmetry and the *XY*-ordered phase at a time $t_c \approx 0.45 t_r$ when $h_c/g_c \approx 1.8(6)$ (Supplementary Information). The transition, analogous to the 2D Mott insulator−superfluid transition[50], is in the universality class of a three-dimensional (3D) *XY* model, with the correlation length and dynamical critical exponents $\nu \approx 0.67$ and $z = 1$, respectively. Following the ramp, we rapidly return back to the idle frequencies within 3 ns and perform measurements of correlation functions.

Figure 3b shows the ramp time dependence of the average energy density, $\varepsilon = n_B^{-1} \sum_{\langle i,j \rangle} \langle X_i X_j + Y_i Y_j \rangle / 2$ averaged over $n_B = 110$ bonds ($N_q = 65$) and corrected for readout errors (Methods). As $t_r$ increases

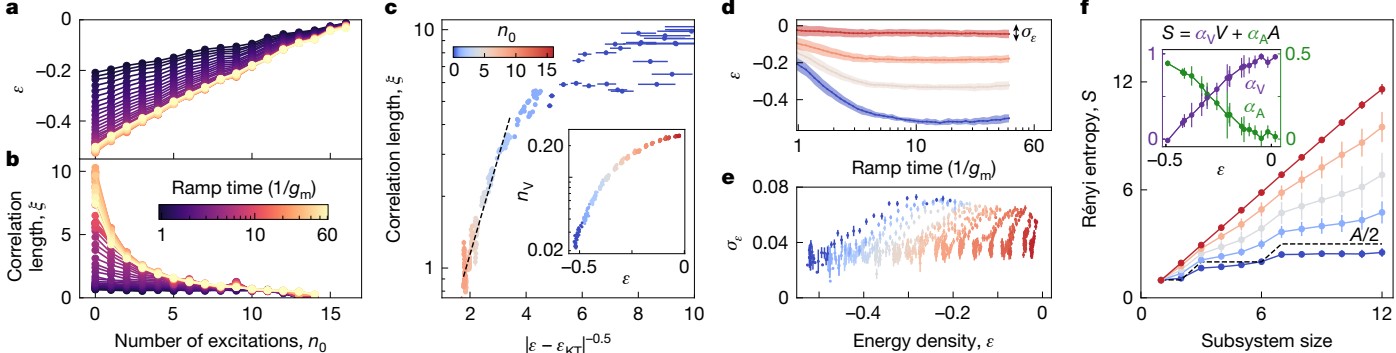

**Fig. 4 | Tunable thermalized states through initial excitations. a,b,** Energy density (**a**) and correlation length (**b**) versus number of initial excitations, $n_0$, averaged over three randomizations. Error bars smaller than markers ($N_{reps}$ = $6 \times 10^3$ per data point). **c,** Energy dependence of $\xi$, demonstrating collapse when data from sweeps of $t_r$ and $n_0$ are plotted together. log $\xi$ is near-linear in $|\varepsilon - \varepsilon_{KT}|^{-0.5}$ as theoretically expected. Inset, vortex density versus energy density, showing similar collapse. Error estimated from bootstrapping in **c**

and **e. d,** $t_r$-dependence of energy density and its fluctuations (width), for various $n_0$. **e,** Energy fluctuations versus energy density, showing an absence of collapse becase $\sigma_\varepsilon$ does not thermalize. **f,** Second Rényi entropy versus subsystem size for various $n_0$ at $t_r$ = 200 ns $\approx 23/g$. Increasing $n_0$ causes transition from area- to volume-law behaviour, also seen from the extracted contributions (inset, error bars represent one standard deviation across five excitation randomizations and four subsystems).

and the dynamics become more adiabatic, we observe a decrease in energy density towards the theoretically predicted ground state value of $\varepsilon_{gs}$ = −0.56, as well as the predicted Kosterlitz–Thouless (KT) transition energy density, $\varepsilon_{KT}$ = −0.53 ± 0.01 (grey and black lines, respectively). As demonstrated below, the final states are thermalized to a strong extent, so $\varepsilon$ can be used to evaluate the final effective temperature. To correct for photon decay errors, we apply digital entangling gates at the end of the circuit to convert each pair of qubits to the Bell basis (Methods). This allows for postselecting with respect to photon number conservation (red squares), which yields an improved value of $\varepsilon$ = −0.53 ± 0.01, roughly equal to the Kosterlitz–Thouless transition point. The remaining discrepancy from $\varepsilon_{gs}$ is attributed to dephasing effects, which are not corrected by this technique.

As the energy itself does not reveal the effects of thermalization, we next turn to correlations at longer distances and consider the average correlation, $\bar{G}(\mathbf{r})$, between pairs of qubits separated by $\mathbf{r}$, shown in Fig. 3c. We observe antiferromagnetic ordering, with the range and magnitude of correlations increasing notably with ramp time, as expected for states with decreasing energy. We next compute the radial average, $\bar{G}(|\mathbf{r}|)$, and fit the resulting decay profiles with exponential fits to extract the correlation length, $\xi$, as well as with power-law fits to evaluate the type of distance-scaling (Fig. 3d). At short ramp times, the correlations are found to decay exponentially, as theoretically expected for states above the Kosterlitz–Thouless transition, in which freely proliferating vortices preclude long-range order. At longer ramp times, on the other hand, the decay behaviour is better described by power-law fits, as shown in Fig. 3e; specifically, we observe a marked decrease in the ratio between the root-mean-square (r.m.s.) errors of power-law and exponential fits to well below 1 near $g_m t_r$ = 25, where the energy is also close to its minimum value. This behaviour is consistent with that expected in the classical critical regime, where free vortices become entropically unfavourable and are replaced by bound vortex–antivortex pairs, leading to algebraically decaying correlations. (We note that finite-size scaling analysis of the Kosterlitz–Thouless transition is challenging, owing to characteristic rapid growth of the correlation length, and is not attempted here.) In the region with good power-law agreement, we extract a power-law exponent of $\gamma$ = −0.29 (Fig. 3f), close to the theoretically expected universal value of $-\frac{1}{4}$ at the Kosterlitz–Thouless transition[51]. To further substantiate our interpretation, we also measure four-qubit correlators to construct the Swendsen proxy for the vortex density[52], given by $n_V = \frac{1}{4N_P} \sum_{i=1}^{N_P} (1 - X_{i1}X_{i3} - Y_{i1}Y_{i3})(1 - X_{i2}X_{i4} - Y_{i2}Y_{i4})$ for plaquettes $i = 1, .., N_P$ with vertices $\{i1, i2, i3, i4\}$. Indeed, we find a rapid decrease

in $n_V$ as $t_r$ is increased (Fig. 3g), to a minimum value of $2 \times 10^{-2}$ in the low-energy regime.

Having studied the classical critical behaviour, we next explore the scaling of the correlation length with the duration $t_r$ over which we sweep through the quantum critical point (Fig. 3h). The correlation length rises to a maximum of $\xi \approx 10$ at $g_m t_r$ = 25, which is equal to the longest dimension of our system. At long ramp times, we observe a slight decrease in $\xi$, attributed to qubit decoherence, as well as periodic oscillations. The latter are also observed in MPS simulations and expected to be caused by finite-size effects. Focusing on shorter ramp times for which these additional effects are absent and the correlations show a more clear exponential decay, we observe strong deviation from the power-law scaling with exponent $v/(1 + vz)$ = 0.4 predicted by KZ theory. Specifically, $\xi$ grows substantially more superlinearly, and clear discrepancies from power-law scaling are observed in both experiment and simulation. We attribute the observed breakdown of KZ scaling to coarsening beyond the expected freezing point[19,21].

To demonstrate this more explicitly, we measure the correlation length along the Hamiltonian ramp (Fig. 3i). The KZ prediction assumes that the dynamics freeze at $t_{KZ}$, when the inverse gap exceeds $|t - t_c|$. By contrast, we find that $\xi$ continues to increase, suggesting that the system is instead able to further thermalize, thus causing a different correlation length at the end of the ramp. To illuminate this point, we plot the experimentally measured correlation lengths at $t_{KZ}$ in Fig. 3h and find better agreement with the KZ prediction. Notably, by rescaling to $\xi/\xi_{KZ}$ and $(t - t_c)/|t_{KZ} - t_c| \equiv \tau/\tau_{KZ}$, we find that the curves collapse to a common $f(\tau/\tau_{KZ})$, consistent with predictions of universal coarsening dynamics[19,21,22]. The collapse extends well beyond the quantum critical regime, $-\tau_{KZ} < \tau < \tau_{KZ}$, indicative of dynamical universality driven by coarsening. We observe behaviour similar to the theoretically predicted $f(x) \propto |x|^{-v}$ for $x < -1$ (small deviations probably related to effects of finite size and short $\xi$), and $f(1)/f(-1)$ = 2.3 ± 0.1. We heuristically find approximately $f(x) \propto x$ for $x > 1$, in which the interplay of gapped and gapless modes is expected to cause different behaviour from quantum Ising models.

Thus far, we have tuned the final energy through the ramp rate of the Hamiltonian. To further study thermalization, as well as the scaling relations near the Kosterlitz–Thouless transition, we prepare a variable number of excitations, $n_0$ (pairs of spin flips in randomized locations) in the initial state[53]. Whereas we find that the final average energy density depends linearly on $n_0$ (Fig. 4a), the behaviour of the correlation length is more intricate (Fig. 4b) and is best understood by plotting $\xi$ versus energy density for all $n_0$ and $g_m t_r > 5$ (Fig. 4c). Notably, the points

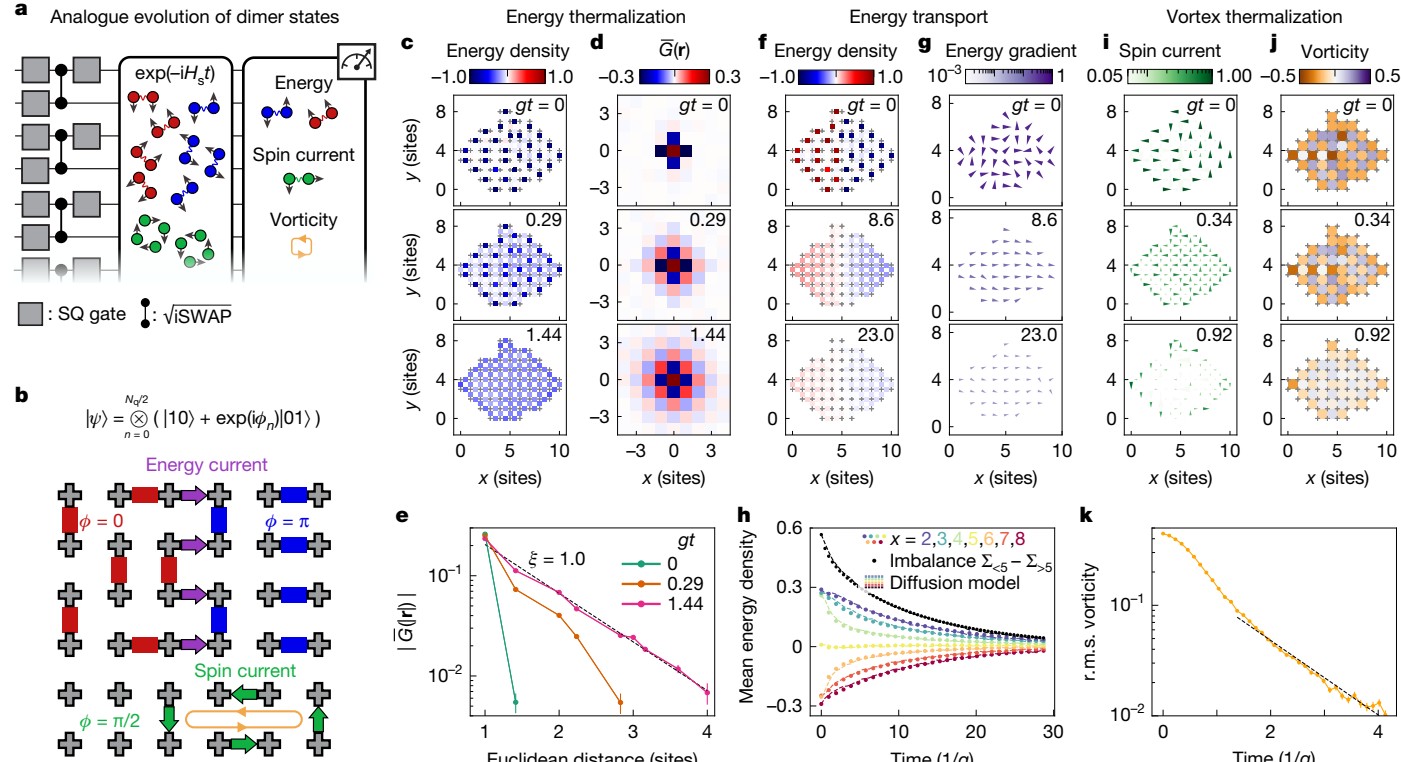

**Fig. 5 | Transport and thermalization dynamics with entangled initial states. a**, Dimer states are prepared using digital gates, and their thermalization and transport dynamics are realized with analogue evolution, before finally measuring energy, spin current and vorticity. **b**, We prepare dimer states with spatially tunable phase, $\phi$. Energy gradients between $\phi = 0$ ($\varepsilon > 0$) and $\phi = \pi$ ($\varepsilon < 0$) drive energy current, whereas $\phi = \pi/2$ gives non-zero spin current and vorticity. **c,d**, Time evolution of energy density (**c**) and correlations (**d**) after dimer preparation demonstrate rapid thermalization. **e**, Correlations become increasingly long-ranged as the system thermalizes. Dashed line, exponential fit. **f,g**, Energy density (**f**) and energy gradients (**g**) after dimer preparation with $\phi = 0$ and $\pi$ in the left and right halves of the system, respectively, showing energy transport on much longer timescales. Colour and length scales of arrows in **g** and **i** are logarithmic. **h**, Time dependence of average energy density along various vertical cuts (coloured circles) and energy imbalance across $x = 5$ (black circles), showing very good agreement with diffusion model (dashed lines). Error bars smaller than markers. **i,j**, Spin current (**i**) and vorticity (**j**) for $\phi = \pi/2$, showing rapid thermalization. **k**, The r.m.s. vorticity shows initial slow dynamics followed by near-exponential decay with rate $\Gamma = 49$ MHz $= 0.85g$ (fit shown by dashed line). Error bars in **e** and **k** are estimated by bootstrapping ($N_{\text{reps}} = 10^4$).

show a collapse (also for $n_V$ in inset), suggesting that the final states are well thermalized, such that the energy density determines $\xi$ and $n_V$, as expected from the eigenstate thermalization hypothesis (ETH)[5,6]. Barring $\xi$ near the system size, we find that $\log \xi$ is nearly linear in $|\varepsilon - \varepsilon_{\text{KT}}|^{-0.5}$, as expected near the Kosterlitz–Thouless transition. This is incompatible with naive KZ scaling, and thus further corroborates its breakdown.

Although thermalization causes states created with different $n_0$ and $t_r$ to have the same observables (for example, $n_V$ and $\xi$) if their final energies are equal, the states themselves are not necessarily the same. This can be seen by studying observables such as the energy fluctuations, $\sigma_\varepsilon = (n_B g_m)^{-1} \sqrt{\langle H_{XY}^2 \rangle - \langle H_{XY} \rangle^2}$ with $H_{XY} = \sum_{\langle i,j \rangle} (X_i X_j + Y_i Y_j)/2$, which trivially commute with $H_{XY}$ and are thus not thermalized under ETH. We next reconstruct $\sigma_\varepsilon$ from two- and four-qubit correlators (Methods) and find that it decreases from 0.07 to 0.02 as we approach the ground state for $n_0 = 0$, whereas its dependence on $n_0$ is weaker (Fig. 4d). The low value of $\sigma_\varepsilon$ compared to the tunable energy range indicates our ability to probe specific parts of the spectrum. Notably, when the full dataset across $t_r$ and $n_0$ is plotted against energy density, the points do not collapse (Fig. 4e). This shows the difference in states accessed by the two tuning techniques, which was previously concealed by the thermalization of $n_V$ and $\xi$.

To further characterize the degree of thermalization, we leverage the fast data acquisition rate of our platform to measure the entanglement entropy for subsystem sizes up to 12 qubits, using randomized measurements[54]. At $n_0 = 0$, we find area-law behaviour (Fig. 4f), which,

up to a subleading logarithmic contribution, is consistent with predictions for low-energy states in the $XY$ model[55]. However, tuning to higher final energies by means of larger $n_0$, we find a continuous crossover to volume-law behaviour (area- and volume-law components in inset), as is expected from ETH for thermalized states at finite energy density[2,36].

We have so far observed signatures of thermalization in the final state of the dynamics, but the thermalization dynamics themselves are still left unexplored. Although we have shown that $t_r$ and $n_0$ are effective for realizing and studying states with a desired energy and energy fluctuations, they are limited when it comes to studying spatiotemporal dynamics; to study a state with substantial correlations ($\langle XX \rangle > 0.1$), a ramp time of more than roughly $1/g$ is required, at which point the system is typically already near equilibrium. Moreover, although these knobs allow for tuning energy density and antiferromagnetic correlations, quantities such as vorticity are out of reach.

Next, we therefore expand the capabilities of our platform by combining the analogue evolution with entangled state preparation by means of high-fidelity (digital) two-qubit gates (Fig. 5a,b). Following the preparation of the dimer state, $(|01\rangle - |10\rangle)^{\otimes N_q/2}$, we rapidly turn on $H_s$ with $g/(2\pi) = 10$ MHz and observe very fast thermalization of the energy density on a timescale of just around $1.5/g$ (Fig. 5c). As the system thermalizes, the range of correlations increases rapidly (Fig. 5d), converging to a correlation length of roughly 1.0 (Fig. 5e). As is expected from ETH, this is in good agreement with $\xi \approx 1.1$ observed for the same energy density ($-0.23g$) when tuning $t_r$ and $n_0$.

Next we leverage the tunability of the phases of the initial dimer states to enable study of transport (Fig. 5b). Specifically, we prepare the dimers in one half of the device in the higher-energy dimer state, $|01\rangle + |10\rangle$ (Fig. 5f). Now the dynamics are found to be substantially slower, with clear spatial non-uniformity remaining even after 23 cycles. We also plot the energy density gradient in Fig. 5g, which quickly establishes a uniform field in the $+x$ direction. Figure 5h shows the time dependence of the average energy density at various vertical cuts (coloured circles), as well as the total energy transfer across $x = 5$ (black circles), which both show excellent agreement with a diffusion model (dashed lines). The energy transport is indeed expected to be diffusive due to the relatively high energy of the dimer state. The data allow for extracting a diffusion constant of $D = 29.6$ MHz $= 0.52g$.

The use of initial entangled states in our hybrid analogue–digital platform enables not only tailoring the initial energy landscape, but also other observables such as vorticity and spin current. We achieve this by further tuning the initial dimer phases to $\pi/2$ (Fig. 5b). This gives rise to local spin currents, $\langle X_i Y_{i+1} - Y_i X_{i+1}\rangle/2 \neq 0$, and a sea of vortices and anti-vortices, quantified by the vorticity, $V_i = \frac{1}{4}(X_{i1}Y_{i2} - Y_{i2}X_{i3} + X_{i3}Y_{i4} - Y_{i4}X_{i1})$ for each plaquette $i$ with vertices $\{i1, i2, i3, i4\}$. The temporal evolution of the spin current and vorticity is presented in Fig. 5i,j, respectively, showing thermalization on a fast timescale similar to that in Fig. 5c. Specifically, after an initial super-exponential decay, the r.m.s. vorticity decays near-exponentially with a rate of $\Gamma = 49$ MHz $= 0.85g$ (Fig. 5k).

Our results demonstrate a high-fidelity quantum simulator with the capability of emulating beyond-classical chaotic dynamics, a wide range of characterization probes and versatile analogue–digital control. Leveraging these features has enabled new insights about the rich interplay of quantum and classical critical behaviour in the 2D $XY$ model, including the Kosterlitz–Thouless transition, thermalization dynamics and a breakdown of the KZ scaling relations. The effects of the co-existing gapped longitudinal modes and gapless (finite-size limited) transverse modes, specifically on the coarsening dynamics, is of particular interest for future theoretical study. Looking ahead, the new platform presented here is expected to offer an invaluable playground for studies of classically intractable many-body quantum physics, including, for example, dynamical response functions and magnetic frustration. Finally, we note that during the preparation of this paper, we became aware of a related work studying coarsening near an Ising quantum phase transition with Rydberg atoms[56].

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

T. I. Andersen[1,16 ✉], N. Astrakhantsev[1,16], A. H. Karamlou[1], J. Berndtsson[1], J. Motruk[2], A. Szasz[1], J. A. Gross[1], A. Schuckert[3], T. Westerhout[4], Y. Zhang[1], E. Forati[1], D. Rossi[2], B. Kobrin[1], A. Di Paolo[1], A. R. Klots[1], I. Drozdov[1,5], V. Kurilovich[1], A. Petukhov[1], L. B. Ioffe[1], A. Elben[6], A. Rath[1], V. Vitale[7], B. Vermersch[7], R. Acharya[1], L. A. Beni[1], K. Anderson[1], M. Ansmann[1], F. Arute[1], K. Arya[1], A. Asfaw[1], J. Atalaya[1], B. Ballard[1], J. C. Bardin[1,8], A. Bengtsson[1], A. Bilmes[1], G. Bortoli[1], A. Bourassa[1], J. Bovaird[1], L. Brill[1], M. Broughton[1], D. A. Browne[1], B. Buchea[1], B. B. Buckley[1], D. A. Buell[1], T. Burger[1], B. Burkett[1], N. Bushnell[1], A. Cabrera[1], J. Campero[1], H.-S. Chang[1], Z. Chen[1], B. Chiaro[1], J. Claes[1], A. Y. Cleland[1], J. Cogan[1], R. Collins[1], P. Conner[1], W. Courtney[1], A. L. Crook[1], S. Das[1], D. M. Debroy[1], L. De Lorenzo[1], A. Del Toro Barba[1], S. Demura[1], P. Donohoe[1], A. Dunsworth[1], C. Earle[1], A. Eickbusch[1], A. M. Elbag[1], M. Elzouka[1], C. Erickson[1], L. Faoro[1], R. Fatemi[1], V. S. Ferreira[1], L. Flores Burgos[1], A. G. Fowler[1], B. Foxen[1], S. Ganjam[1], R. Gasca[1], W. Giang[1], C. Gidney[1], D. Gilboa[1], M. Giustina[1], R. Gosula[1], A. Grajales Dau[1], D. Graumann[1], A. Greene[1], S. Habegger[1], M. C. Hamilton[1,9], M. Hansen[1], M. P. Harrigan[1], S. D. Harrington[1], S. Heslin[1,9], P. Heu[1], G. Hill[1], M. R. Hoffmann[1], H.-Y. Huang[1], T. Huang[1], A. Huff[1], W. J. Huggins[1], S. V. Isakov[1], E. Jeffrey[1], Z. Jiang[1], C. Jones[1], S. Jordan[1], C. Joshi[1], P. Juhas[1], D. Kafri[1], H. Kang[1], K. Kechedzhi[1], T. Khaire[1], T. Khattar[1], M. Khezri[1], M. Kieferová[1,10], S. Kim[1], A. Kitaev[1], P. Klimov[1], A. N. Korotkov[1,11], F. Kostritsa[1], J. M. Kreikebaum[1], D. Landhuis[1], B. W. Langley[1], P. Laptev[1], K.-M. Lau[1], L. Le Guevel[1], J. Ledford[1], J. Lee[1,12], K. W. Lee[1], Y. D. Lensky[1], B. J. Lester[1], W. Y. Li[1], A. T. Lill[1], W. Liu[1], W. P. Livingston[1], A. Locharla[1], D. Lundahl[1], A. Lunt[1], S. Madhuk[1], A. Maloney[1], S. Mandrà[1], L. S. Martin[1], O. Martin[1], S. Martin[1], C. Maxfield[1], J. R. McClean[1], M. McEwen[1], S. Meeks[1], K. C. Miao[1], A. Mieszala[1], S. Molina[1], S. Montazeri[1], A. Morvan[1], R. Movassagh[1], C. Neill[1], A. Nersisyan[1], M. Newman[1], A. Nguyen[1], M. Nguyen[1], C.-H. Ni[1], M. Y. Niu[1], W. D. Oliver[1], K. Ottosson[1], A. Pizzuto[1], R. Potter[1], O. Pritchard[1], L. P. Pryadko[1,11], C. Quintana[1], M. J. Reagor[1], D. M. Rhodes[1], G. Roberts[1], C. Rocque[1], E. Rosenberg[1], N. C. Rubin[1], N. Saei[1], K. Sankaragomathi[1], K. J. Satzinger[1], H. F. Schurkus[1], C. Schuster[1], M. J. Shearn[1], A. Shorter[1], N. Shutty[1], V. Shvarts[1], V. Sivak[1], J. Skruzny[1], S. Small[1], W. Clarke Smith[1], S. Springer[1], G. Sterling[1], J. Suchard[1], M. Szalay[1], A. Sztein[1], D. Thor[1], A. Torres[1], M. M. Torunbalci[1], A. Vaishnav[1], S. Vdovichev[1], B. Villalonga[1], C. Vollgraff Heidweiller[1], S. Waltman[1], S. X. Wang[1], T. White[1], K. Wong[1], B. W. K. Woo[1], C. Xing[1], Z. Jamie Yao[1], P. Yeh[1], B. Ying[1], J. Yoo[1], N. Yosri[1], G. Young[1], A. Zalcman[1], N. Zhu[1], N. Zobrist[1], H. Neven[1], R. Babbush[1], S. Boixo[1], J. Hilton[1], E. Lucero[1], A. Megrant[1], J. Kelly[1], Y. Chen[1], V. Smelyanskiy[1], G. Vidal[1], P. Roushan[1], A. M. Läuchli[13,14], D. A. Abanin[1,15 ✉] & X. Mi[1 ✉]

[1]Google Research, Mountain View, CA, USA. [2]Department of Theoretical Physics, University of Geneva, Geneva, Switzerland. [3]Joint Quantum Institute and Joint Center for Quantum Information and Computer Science, NIST/University of Maryland, College Park, MD, USA. [4]Institute of Molecules and Materials, Radboud University, Nijmegen, The Netherlands. [5]Department of Physics, University of Connecticut, Storrs, CT, USA. [6]Institute for Quantum Information and Matter and Walter Burke Institute for Theoretical Physics, Caltech, Pasadena, CA, USA. [7]Université Grenoble Alpes, CNRS, LPMMC, Grenoble, France. [8]Department of Electrical and Computer Engineering, University of Massachusetts, Amherst, MA, USA. [9]Department of Electrical and Computer Engineering, Auburn University, Auburn, AL, USA. [10]QSI, Faculty of Engineering and Information Technology, University of Technology Sydney, Sydney, New South Wales, Australia. [11]Department of Electrical and Computer Engineering, University of California, Riverside, Riverside, CA, USA. [12]Department of Chemistry and Chemical Biology, Harvard University, Cambridge, MA, USA. [13]Laboratory for Theoretical and Computational Physics, Paul Scherrer Institute, Villigen, Switzerland. [14]Institute of Physics, Ecole Polytechnique Fédérale de Lausanne (EPFL), Lausanne, Switzerland. [15]Department of Physics, Princeton University, Princeton, NJ, USA. [16]These authors contributed equally: T. I. Andersen, N. Astrakhantsev. ✉e-mail: trondiandersen@google.com; abanin@google.com; mixiao@google.com

## Methods

### Device details

The experiments are performed on a superconducting quantum processor with frequency-tunable transmon qubits and couplers, with a similar design to that in ref. 45. Extended Data Fig. 1a,b show the measured Ramsey dephasing ($T_2^*$) and photon relaxation ($T_1$) times at the interaction frequency of 5.93 GHz used in our experiments, with median values of 2.0 and 18.8 µs, respectively. Characterizing our digital gate performance, we find a median Pauli error of $4.5 \times 10^{-3}$ for combined $\sqrt{\text{iSWAP}}$ and single-qubit gates (Extended Data Fig. 1c), and $1.0 \times 10^{-3}$ for single-qubit gates alone (Extended Data Fig. 1d). Finally, Extended Data Fig. 1e shows our readout errors, with a median of $1.4 \times 10^{-2}$.

### Analogue calibration

In this section, we describe our new, scalable analogue calibration framework that enables roughly 0.1% cycle error per qubit. To achieve a scalable scheme, we perform pairwise calibration measurements—specifically single-photon and swap spectroscopy—which allows for accurately setting the effective coupling $\widetilde{g}$ and dressed qubit frequencies $\widetilde{\omega}_{qi}$ in each qubit pair. A key challenge in analogue calibration that contrasts with its digital counterpart is that these dressed quantities in the pairwise scenario change drastically when all couplers are turned on in the fully coupled global case. Therefore, we perform extensive modelling of the device physics to accurately convert them to the bare qubit and coupler frequencies, $\{\omega_{qi}\}$, $\{\omega_{cj}\}$, which, crucially, do not change from the local calibration measurements to the full-scale experiments.

**Model device Hamiltonian.** We model both the qubits and couplers in our tunable coupler architecture as Kerr oscillators, with four or five levels in each transmon, depending on the number of photons involved in the Hamiltonian term of interest. Specifically, in calculations concerning one- and two-photon terms, we include four and five levels, respectively. This is done to account for effects that do not obey the rotating-wave approximation, which couple $|1\rangle$ to $|3\rangle$ and $|2\rangle$ to $|4\rangle$. To ensure high accuracy, we account for not only coupling terms between neighbouring qubits and couplers, but also diagonal pathways, including between couplers:

$$
H_{\mathrm{d}} = \overbrace{\sum_{qi} \omega_{qi}\hat{n}_{qi} - \eta_{qi}\hat{n}_{qi}(\hat{n}_{qi}-1)/2}^{\text{Single qubit}}
$$
$$
+ \overbrace{\sum_{cj} \omega_{cj}\hat{n}_{cj} - \eta_{cj}\hat{n}_{cj}(\hat{n}_{cj}-1)/2}^{\text{Single coupler}}
$$
$$
+ \overbrace{\sum_{qi,qj} \tfrac{1}{2}\widetilde{k}_{qi,qj}\sqrt{\omega_{qi}\omega_{qj}}\,\widehat{Q}_{qi}\widehat{Q}_{qj}}^{\text{Qubit–qubit coupling}} \quad (2)
$$
$$
+ \overbrace{\sum_{qi,cj} \tfrac{1}{2}\widetilde{k}_{qi,cj}\sqrt{\omega_{qi}\omega_{cj}}\,\widehat{Q}_{qi}\widehat{Q}_{cj}}^{\text{Qubit–coupler coupling}}
$$
$$
+ \overbrace{\sum_{ci,cj} \tfrac{1}{2}\widetilde{k}_{ci,cj}\sqrt{\omega_{ci}\omega_{cj}}\,\widehat{Q}_{ci}\widehat{Q}_{cj}}^{\text{Coupler–coupler coupling}},
$$

where $\widehat{Q} = a^\dagger + a$ and the $\widetilde{k}$ are the effective coupling efficiencies between transmons, including both direct and indirect capacitive contributions (note that the indirect contributions should not be confused with contributions due to virtual exchange interactions, which are included indirectly when we project out the couplers later on). The coupling efficiencies for the various terms can be summarized as follows:

For $k_{\mathrm{qq}}$, we include three types of qubit–qubit coupling, distinguished by the relative positioning of the qubits. Notably, the geometry of the transmons breaks the 90° rotational symmetry; specifically, the couplings differ along the northwest–southeast (NW–SE) and northeast-southwest (NE–SW) directions. To discuss the three types of coupling, we consider the four qubits on a plaquette shown in Extended Data Fig. 2 and consider examples of pairs of transmons (the formulas for the remaining pairs are given by reflection symmetry about the NW–SE and NE–SW axes, for example, $\widetilde{k}_{q_1,c_{23}} = k_{q_1,c_{23}} + 2k_{q_1,q_2}k_{q_2,c_{23}}$ infers that $\widetilde{k}_{q_1,c_{34}} = k_{q_1,c_{34}} + 2k_{q_1,q_4}k_{q_4,c_{34}}$):

(1) Nearest-neighbours qubits, $q_1$ and $q_2$ separated by a coupler $c_{12}$:
$\widetilde{k}_{q_1,q_2} = k_{q_1,q_2} + k_{q_1,c}k_{q_2,c}$.
(2) Diagonally separated qubits in the NW–SE direction, $q_1$ and $q_3$:
$\widetilde{k}_{q_1,q_3} = k_{q_1,q_3} + 2(k_{q_1,q_2}k_{q_2,q_3} + k_{q_1,q_4}k_{q_4,q_3})$.
(3) Diagonally separated qubits in the NE–SW direction, $q_2$ and $q_4$:
$\widetilde{k}_{q_2,q_4} = k_{q_2,q_4}$.

For $k_{\mathrm{qc}}$, we also include three types of qubit–coupler coupling:
(1) Nearest-neighbours: $\widetilde{k}_{q_1,c_1} = k_{q_1,c_1}$.
(2) Diagonally separated qubit and coupler in the NW–SE direction, $q_1$ and $c_{23}$: $\widetilde{k}_{q_1,c_{23}} = k_{q_1,c_{23}} + 2k_{q_1,q_2}k_{q_2,c_{23}}$.
(3) Diagonally separated qubit and coupler in the NE–SW direction, $q_4$ and $c_{12}$: $\widetilde{k}_{q_4,c_{12}} = k_{q_4,c_{12}}$.

For $k_{\mathrm{cc}}$, we consider two types of coupler–coupler coupling:
(1) Diagonally separated couplers in the NW–SE direction $c_{12}$ and $c_{23}$: $\widetilde{k}_{c_{12},c_{23}} = k_{c_{12},c_{23}} + 2k_{c_{12},q_2}k_{q_2,c_{23}}$.
(2) Diagonally separated qubit and coupler in the NE–SW direction, $c_{12}$ and $c_{14}$: $\widetilde{k}_{c_{12},c_{14}} = k_{c_{12},c_{14}}$.

**Calibration experiments.** To calibrate the bare qubit and coupler frequencies for a given set of applied biases, we perform various types of calibration measurements (Extended Data Fig. 3a):

**Ramsey spectroscopy.** In this measurement, we perform standard Ramsey spectroscopy for a range of applied qubit bias values, while keeping the couplers turned off and the neighbouring qubits detuned, to prevent swapping.

**Swap spectroscopy.** This measurement is performed on a pairwise level, in which neighbouring couplers (except the one connecting the pair) are turned off. The two qubits are prepared in the $|10\rangle$-state and we measure the swap rate as a function of detuning between the two qubits (Extended Data Fig. 3b). The minimum swap rate tells us the effective coupling between the two qubits, $\widetilde{g}$, and the detuning at which this occurs equals the difference between the dressed frequencies of the qubits, $\widetilde{\omega}_{q_1} - \widetilde{\omega}_{q_2}$ (Extended Data Fig. 3c). Using an iterative scheme, we calibrate the coupler bias required to achieve the target effective coupling.

**Single-photon spectroscopy.** Whereas the swap spectroscopy provides us with the difference of the dressed frequencies, we also need to find their sum to determine the individual values, $\widetilde{\omega}_{q_1}$ and $\widetilde{\omega}_{q_2}$. We achieve this by preparing the qubits in $(|1\rangle + |0\rangle)|0\rangle/\sqrt{2}$ and measuring $\langle X + iY \rangle$ as a function of evolution time (Extended Data Fig. 3d). The Fourier transform of the signal then reveals the eigenfrequencies of the two-qubit system, the average of which is equal to $(\widetilde{\omega}_{q_1} + \widetilde{\omega}_{q_2})/2$ (Extended Data Fig. 3e).

Next, using separately calibrated coupling efficiencies, we model all the calibration experiments above with the device Hamiltonian described earlier, to find the bare qubit and coupler frequencies that give the dressed quantities observed in the calibration experiment. We model not only the two qubits and the coupler involved in pairwise experiments (single qubit involved in Ramsey), but also the neighbouring 'padding' qubits and couplers to account for their effects. Therefore, we start by determining the bare idle frequencies, $\{\omega^{\mathrm{idle}}\}$, because these must be known to represent the padding in the interaction configuration.

**Projection onto computational subspace.** Considering the fact that our model device Hamiltonian involves both qubits and couplers with up to five levels in each, it is computationally intractable to use it for time evolution even at small photon numbers. Moreover, in this form,

it is very difficult to map its behaviour onto physically relevant systems. We therefore perform a projection technique to convert the device Hamiltonian into a spin Hamiltonian, $H_s$, that acts on the computational subspace. To find spin Hamiltonian terms involving $n$ photons in a system of $N_q$ qubits, we write $H^{(n)} = \sum_{i,j} |i\rangle\langle i|H_d|j\rangle\langle j|$, where $\{|i\rangle\}$ are our $N_n = \binom{N_q}{n}$ new dressed $n$-photon basis states.

Let us now motivate our choice of dressed basis states, by considering a few different options. One option could have been to simply use the bare qubit states, $\{|i\rangle_{bare}\}$; however, this would cause the spin Hamiltonian to have different eigen-energies from the low-energy spectrum of $H_d$. A second option would be to instead use the $N_n$ lowest-energy $n$-photon eigenstates of $H_d$, $\{|i\rangle_{eigen}\}$. In this case, the spin Hamiltonian is guaranteed to have the same $N_n$ lowest-energy $n$-photon eigen-energies as $H_d$. However, these basis states are highly delocalized and poorly represent our qubits. Hence, to get the best of both worlds, we turn to a third option, in which we project the bare qubit states onto the low-energy eigenspace spanned by $\{|i\rangle_{eigen}\}$. These projections are not orthonormal, so we perform singular value decomposition and set the singular values to one to arrive at our new dressed basis states. It can be shown that this is the most localized set of states that still preserve the low-energy eigenvalues[57]. These new basis states are slightly delocalized on the nearest couplers and qubits, and also have a weak overlap with states that have $n + 2$ and $n - 2$ photons due to terms beyond the rotating-wave approximation. We note that our typical coupler ramp times of more than 5 ns are sufficient to ensure adiabatic conversion between the bare qubit states (in which we perform state preparation and measurement) and the dressed basis states that are relevant under analogue evolution.

The spin Hamiltonian $H^{(n)}$ found from the technique above in principle includes all terms involving $\leq n$ photons, including very long-range interactions; however, they drop off rapidly with the photon–photon separation $d$ (typically as $(g/\eta)^d \sim 0.1^d$). Moreover, we also find that the terms decay with the number of involved photons in a similar way. Hence, to achieve the low error demonstrated in our manuscript, it is sufficient to include only terms involving up to two photons, and where all the involved qubits are a maximum Manhattan distance of two sites apart, resulting in:

$$
\begin{aligned}
H = \sum_i \omega_i n_i &+ \sum_{\langle i, j \rangle} g_{ij}(X_iX_j + Y_iY_j)/2 + \sum_{\langle i, j \rangle} g_{ij}^{nn} n_i n_j \\
&+ \sum_{\langle i, j, k \rangle} (g_{ijk}^{XnX} n_j + g_{ijk}^{XIX})(X_iX_k + Y_iY_k)/2 \\
&+ \sum_{\langle i, j, k \rangle} (g_{ijk}^{nXX} n_i(X_jX_k + Y_jY_k)/2,
\end{aligned}
\tag{3}
$$

where $g_{ij}^{nn}$, $g_{ijk}^{XnX}$ and $g_{ijk}^{XIX}$ scale as $g^2/\eta$, while $g_{ijk}^{nXX}$ scales as $g^3/\eta^2$ and qubits $i, j, k$ are connected (Extended Data Fig. 4). We note that there is an offset to these scaling behaviours, which arises due to the diagonal capacitive coupling. This is particularly evident for terms involving qubits along the NW–SE diagonal, because the diagonal coupling is strongest there.

Our technique requires finding the $N_n$ lowest-energy $n$-photon eigenstates of $H_d$, which has a high computational cost for large $N_q$. Fortunately, for a given Hamiltonian term involving a certain set of qubits, the effect of other transmons decays quickly with distance, and we only need to include the nearest neighbouring qubits and couplers to achieve accuracies on the tens of kHz scale. To find the spin Hamiltonian terms, we therefore scan through various subsystems and perform the procedure outlined above for each of them.

## Phase calibration for hybrid analogue–digital experiments
In experiments in which we prepare an entangled initial state, the frequency trajectories of the qubits lead to phase accumulation that must be characterized and corrected through phase gates, both before

and after the analogue evolution (Extended Data Fig. 5a). Specifically, in the frame that rotates at the interaction frequency, the qubits in each dimer pair precess relative to each other before they reach the interaction frequency. Hence, a phase rotation $\phi_{0,i}$ must be applied to every qubit before turning on the analogue Hamiltonian to ensure that the dimer pairs have the desired phase difference when the coupling is turned on. Second, in the idle frame (in which we perform the final measurements) the qubits are precessing relative to each other while on resonance. Hence, a final phase correction $\phi_{1,i} + \omega_i t$ (where $t$ is the analogue evolution time) must also be applied to every qubit before measurements. These corrections are very sensitive to timing and dispersive shifts: before the analogue evolution, a timing delay in dimer generation of only 150 ps corresponds to a 0.1-rad change in $\phi_0$ for an idle frequency difference of 100 MHz. Furthermore, during the idle evolution, a 0.1% (80 kHz) change in dispersive shift leads to a 0.1-rad change in the final phase after 200 ns of analogue evolution. Hence, standard calibration techniques, such as single-qubit Ramsey spectroscopy, in which the configuration is sufficiently different from that in the actual experiment, are not accurate enough. We therefore use a set of three calibration techniques for $\phi_{0,i}$, $\phi_{1,i}$ and $\omega_i$ that are designed to represent the configuration used in the actual experiment as well as possible:

To calibrate $\phi_{0,i}$, we make use of the fact that the dimer state is only an eigenstate of the coupling Hamiltonian when the phase difference of the qubits is 0 or $\pi$. Hence, we sweep the phase difference and measure the population oscillations between the qubits with time. The correct phase compensation is the one that minimizes the amplitude of the population oscillations. We note two important points about this calibration step: first, as the measurements are in the $Z$-basis, they do not depend on the calibration of $\phi_{1,i}$ and $\omega_i$. Second, because the phase calibrated in this step is accumulated before the couplers are turned on, it is not affected by dispersive shifts. It is therefore not a problem that neighbouring couplers are turned off during this particular step.

As mentioned previously, the calibration of $\omega_i$ is very sensitive to dispersive shifts and must therefore be performed in the exact same configuration as the actual experiment. We achieve this by performing the KZ experiment (ramp from Neel state in staggered field) with a slow ramp and leaving the analogue Hamiltonian on for a variable time (Extended Data Fig. 5d). The resultant state shows long-range $XX + YY$ correlations, and the effect of the phase accumulation in the idle frame is to cause oscillations in the correlator between each pair $i$ and $j$ with a frequency $\omega_i - \omega_j$ (Extended Data Fig. 5e). Hence, by measuring the frequency of oscillations of all the correlators, the full set of $\{\omega_i\}$ can be determined. The key advantage of this calibration measurement is that all the couplers are turned on, so that the dispersive shifts are the same as in the actual dimer experiment. However, the initial part of the KZ circuit—including the initial staggered field and the slow ramp of the couplers—is different, so the time-independent part of the phase correction, $\phi_{1,i}$, must be calibrated separately.

Finally, to determine $\phi_{1,i}$, we take advantage of energy conservation. Specifically, we perform the dimer experiment with single dimers while sweeping their final phase difference (Extended Data Fig. 5f). Only the correct phase compensation leads to $\langle X_1X_2 \rangle = 1$ and conserved energy, as can be see in Extended Data Fig. 5g. Whereas the dispersive shifts from neighbouring couplers affect the time-dependent part of the final phase $\omega_i t$ and thus had to be included in the previous step, they do not have this effect on $\phi_{1,i}$ and can therefore be excluded here.

Finally, we note that for experiments not involving entangled initial states (Figs. 3 and 4), only the step for calibration of $\{\omega_i\}$ outlined above is required.

## Readout correction and postselection schemes
**Bell measurements.** When measuring $\langle XX + YY \rangle$ correlators using standard single-qubit measurements, we cannot simultaneously get information about the number of photons measured on the pair of

qubits, preventing us from postselecting our data on photon conservation. To get around this for nearest-neighbour pairs, we change our measurement basis by applying an entangling gate given by the unitary,

$$\begin{bmatrix} 1 & 0 & 0 & 0 \\ 0 & 1/\sqrt{2} & -1/\sqrt{2} & 0 \\ 0 & 1/\sqrt{2} & 1/\sqrt{2} & 0 \\ 0 & 0 & 0 & 1 \end{bmatrix}$$

to each pair. From these measurements, we can deduce both the nearest-neighbour correlators and the number of photons present. We use this technique to process the data labelled 'Bell' in Fig. 3b. We find good alignment between direct measurements of the correlators and the inferred correlators from the Bell measurements.

**Bell measurements with readout corrections.** Typically, one can correct for readout errors by inverting the error channel. In the case in which readout errors are uncorrelated, we can simply characterize the matrix $\beta$ for each qubit

$$\beta = \begin{bmatrix} p_{(0|0)} & p_{(0|1)} \\ p_{(1|0)} & p_{(1|1)} \end{bmatrix}$$

where $p_{(i|j)}$ is the probability of measuring a state $|i\rangle$ given that $|j\rangle$ was prepared[58]. In the case in which readout errors are correlated for pairs, we can similarly characterize a matrix $\gamma$ for each pair

$$\gamma = \begin{bmatrix} p_{(00|00)} & p_{(00|01)} & p_{(00|10)} & p_{(00|11)} \\ p_{(01|00)} & p_{(01|01)} & p_{(01|10)} & p_{(01|11)} \\ p_{(10|00)} & p_{(10|01)} & p_{(10|10)} & p_{(10|11)} \\ p_{(11|00)} & p_{(11|01)} & p_{(11|01)} & p_{(11|11)} \end{bmatrix}$$

where $p_{(ij|ab)}$ is the probability of measuring a state $|ij\rangle$ given that $|ab\rangle$ was prepared. One can compensate for the effects of readout errors on an observable by inverting these matrices and applying them to the measured distribution of bitstrings of the subsystem involved in the observable.

In a case in which we want to both correct for readout errors and postselect our data, we cannot apply the readout correction on the postselected distributions as this would overcorrect for $p_{(0|1)}$ type errors. We also cannot simply correct the distributions of subsystem bitstrings before the postselection process because we need access to the global bitstrings to postselect on photon number conservation. Instead, we use a Markov-like process in which we consider each individual bitstring, and flip pairs of spins according to the probabilities inferred from the $\gamma$ matrices. We then postselect the individual bitstrings on the criteria of photon conservation and, finally, compute the quantity of interest.

To confirm the validity of this method, we classically simulate a low-temperature state of the $XY$ model for 64 qubits (using the ground state of two disconnected sets of 32 qubits), introduce noise to the system and use the above protocol to correct for the $T_1$ and readout errors. In simulating the readout errors, we include a readout bias equal to that observed in experiment, namely $p_{(0|1)}/p_{(1|0)} = 3.7$. We compute the energies of the system after various correction schemes and compare to the noiseless value. The results from these simulations are shown in Extended Data Fig. 6a,b, where we evaluate the performance for a wide range of readout error and probability of photon decay, respectively. The combined technique described above is found to provide the most accurate estimate of the actual energy across a very wide parameter range, extending beyond the range relevant to our experiment (in the experiment, we have readout errors in the range 1–4% and a probability of photon decay of 3–6% for ramp times of 200–500 ns). For very high $T_1$ errors, we find that the error in the

combined technique eventually becomes slightly higher than that of pure postselection. In the special case of very low $T_1$ errors, we observe an interesting effect that leads to a slight underestimate of the energy, which can be understood as follows. Whereas the stochastic compensation of readout errors perfectly re-establishes the correct distributions of subsystem bitstrings (by construction of the probabilities with which we change the two-qubit bitstrings), each individual global bitstring has a non-zero probability of having the wrong total number of photons, even in the case of zero T1 error. The lowest-energy two-qubit state, $|10\rangle - |01\rangle$ (converted to $|10\rangle$ by Bell conversion) has a slightly higher chance of being postselected than other two-qubit states. The result of this is a slight underestimate of the energy, which we emphasize is very small (roughly 1%) and not relevant in the parameter range of our experiment.

## Comparison of ⟨XX⟩ and ⟨YY⟩

The final states produced after the ramp procedures in Figs. 3 and 4 are expected to be $U(1)$-symmetric, and thus have equally strong $XX$ and $YY$ correlations. We here check this by comparing $\langle XX \rangle$ and $\langle YY \rangle$ averaged over all nearest-neighbour qubit pairs across a range of ramp times (Extended Data Fig. 7), and indeed find that the two are equal.

## Diffusion model

In Fig. 5h, we fit the observed energy transport with a diffusion model, which we describe in further detail here. We define the energy density at site $(i,j)$, $e_{i,j}(t)$, as the average of the energy ($\langle XX + YY \rangle/2$) on the bonds that include site $(i,j)$ and model the transport using a simple discretized version of the diffusion equation:

$$\frac{de_{i,j}}{dt} = D(e_{i+1,j} + e_{i-1,j} + e_{i,j-1} + e_{i,j+1} - 4e_{i,j}) \tag{4}$$

where the diffusion constant, $D$, is the only fit parameter.

## Measurements of energy density fluctuations

We use measurements of two- and four-qubit correlators to reconstruct the energy density fluctuations, $\sigma_\varepsilon = (n_B g_m)^{-1} \sqrt{\langle H_{XY}^2 \rangle - \langle H_{XY} \rangle^2}$, with:

$$\frac{H_{XY}^2}{g_m^2} = \left( \sum_{\langle i,j \rangle} (X_i X_j + Y_i Y_j)/2 \right)^2 = \sum_{\langle i,j \rangle} (1 - Z_i Z_j)/2$$
$$+ \sum_{\langle i,j \rangle} \sum_{\langle m,n \rangle} (X_i X_j X_m X_n + Y_i Y_j Y_m Y_n + X_i X_j Y_m Y_n +$$
$$+ Y_i Y_j X_m X_n)/4 + \sum_{\langle i,j \rangle, \langle j,k \rangle} (X_i X_k + Y_i Y_k)/2, \tag{5}$$

where $\langle i,j \rangle$, $\langle j,k \rangle$ and $\langle m,n \rangle$ are nearest-neighbour pairs and $i,j,k,m,n$ are distinct (note that $j$ is included in the last sum to count the number of length-2 paths from $i$ to $k$). Almost all of these terms can be reconstructed from just three different sets of measurements, namely $\{X_i\}$, $\{Y_i\}$ and $\{Z_i\}$, except the four-qubit correlators involving both $X$ and $Y$. To determine these, we measure eight periodic patterns of $X, Y$ shown in Extended Data Fig. 8a, and leverage the substantial degree of isotropy to find the remaining correlators not included in these patterns (further justification below). As shown in Extended Data Fig. 8b, the four-qubit correlators that involve both $X$ and $Y$ show a clear trend with the distance between the centres of mass of the two involved nearest-neighbour pairs $(i,j)$ and $(m,n)$, and we therefore interpolate the data obtained from these eight sets of measurements to find the remaining terms. Determining $\sigma_\varepsilon$ with good relative accuracy is challenging, owing to the very small relative difference between $\langle H_{XY} \rangle^2$ and $\langle H_{XY}^2 \rangle$. Nevertheless, we find that our technique works well, and we obtain relatively good agreement with MPS simulations (Extended Data Fig. 8c).

To further justify the use of this interpolation technique, we show the dependence of $\langle X_i X_j Y_m Y_n \rangle$ on the relative position of the centres of mass of the two involved nearest-neighbour pairs $(i,j)$ and $(m,n)$, showing near-isotropic distributions (Extended Data Fig. 9a). We observe a weak angular dependence with a period of $\pi$ (Extended Data Fig. 9b), which becomes most pronounced when the correlation length is maximized (for example, $g_m t_r = 12.3$). The amplitude is only roughly $\pm 0.01$ (or roughly 5% of the signal itself) and is expected to be due to the system shape. As we are only interested in the sum of all the correlators, this small degree of isotropy has very little effect on the interpolation scheme described above. In Extended Data Fig. 9c, we compare the result of radial interpolation of $\langle XXYY \rangle$ at distance 5 (dashed black curve) to the actual correlators (coloured circles in main) and their average (red dashed curve in inset), and find that the difference is very small. In particular, we quantify the relative difference between the radial interpolation and the averaged actual correlators in Extended Data Fig. 9d, and find that it is on the order of a few percent, and even smaller at the long times that are most essential to our conclusions. These deviations are comparable to the statistical noise (as shown by the error bars) and do not contribute a dominant effect to the total energy fluctuations.

## Correlation fitting

We here provide further details about the fitting procedures used in the main text for analysing correlations. As shown in Extended Data Fig. 10 and also in some of the curves in Fig. 3d, we observe distortions in the correlation decay at longer distances both in experiment (a) and simulation (b), which are expected to be due to the finite size of our system. Specifically, we find that the correlations drop rapidly for some ramp times and start increasing at others. If fitting up to the longest distances, these effects have a strong impact on the analysis, as can be seen from the sharp upturn in the fit error as we exceed a fit range of roughly six sites in Extended Data Fig. 10c. Informed by these findings, and the fact that the maximum distance at which such effects are still minimal is six sites, we use this as the fit-range cut-off. Note that we also observe a noise floor in the correlations around $10^{-2}$, and we therefore do not fit data points smaller than this value.

We investigate the dependence on fit range further by plotting the r.m.s. fit errors for all ramp times and a wide range of fit-range cut-offs in Extended Data Fig. 10d,e (power-law and exponential fits, respectively). From these plots, it is again evident that the fits with distance cut-offs longer than six sites have particularly high errors (it is of course natural to see some increase in error with increasing fit range, but we are here referring to the distinct increase seen especially well in the inset of Extended Data Fig. 10d). Plotting the error ratio in Extended Data Fig. 10f, we find that all fits up to a fit range of seven sites show the same drop below one around $g_m t_r = 10$, and the discrepancy from KZ scaling is observed for all fit ranges (Extended Data Fig. 10g).

## Data availability

The data that support the findings in this study are available at Zenodo (https://doi.org/10.5281/zenodo.14060446)[59].

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

**Acknowledgements** We acknowledge useful discussions with R. Samajdar, D. A. Huse and S. Choi. A. Schuckert acknowledges support from the US Department of Energy, Office of Science, National Quantum Information Science Research Centers, Quantum Systems Accelerator. J.M. acknowledges funding through SNSF Swiss Postdoctoral Fellowship, grant no. 210478. A.E. acknowledges funding by the German National Academy of Sciences Leopoldina under the grant number LPDS 2021-02 and by the Walter Burke Institute for Theoretical Physics at Caltech. Work in Grenoble is funded by the French National Research Agency through the JCJC project QRand (grant no. ANR-20-CE47-0005), Laboratoire d'excellence LANEF (grant no. ANR-10-LABX-51-01), from the Grenoble Nanoscience Foundation.

**Author contributions** T.I.A., D.A.A. and X.M. conceived the project and designed the experiments. T.I.A., A.H.K. and J. Berndtsson performed the experiments and data analysis (A.H.K., entanglement entropy measurements and J. Berndtsson, measurements of ramp time dependence). T.I.A., J.A.G., X.M. and D.A.A. developed the calibration procedures, with assistance from N.A., Y.Z., E.F., B.K., A D.P., A.R.K., I.D., A. Petukhov and L.B.I. J.M., A. Szasz, D.R. and D.A.A. performed MPS simulations and theoretical work with A.M.L. N.A. performed XEB analysis, classical complexity estimates and exact state vector simulations with assistance from T. Westerhout, V.K. and A. Szasz. A. Elben, A.R., V.V. and B. Vermersch performed theoretical work on randomized measurements. T.I.A., N.A., A.M.L., D.A.A. and X.M. wrote the manuscript. D.A.A. and X.M. led and coordinated the project. T.I.A., N.A., A.H.K., J. Berndtsson, J.M., A. Szasz, J.A.G., A. Schuckert, T. Westerhout, Y.Z., E.F., D.R., B.K., A.D.P., A.R.K., I.D., V.K., A. Petukhov, L.B.I., A. Elben, A.R., V.V., B. Vermersch, R.A., L.A.B., K. Anderson, M. Ansmann, F.A., K. Arya, A.A., J.A., B. Ballard, J.C.B., A. Bengtsson, A. Bilmes, G.B., A. Bourassa, J. Bovaird, L.B., M.B., D. A. Browne, B. Buchea, B.B.B., D. A. Buell, T.B., B. Burkett, N.B., A. Cabrera, J. Campero, H.-S.C., Z.C., B.C., J. Claes, A.Y.C., J. Cogan, R.C., P.C., W.C., A.L.C., S. Das, D.M.D., L.D.L., A.D.T.B., S. Demura, P.D., A.D., C. Earle, A. Eickbusch, A.M.E., M.E., C. Erickson, L.F., R.F., V.S.F., L.F.B., A.G.F., B.F., S.G., R. Gasca, W.G., C.G., D. Gilboa, M.G., R. Gosula, A.G.D., D. Graumann, A.G., S. Habegger, M.C.H., M.H., M.P.H., S.D.H., S. Heslin, P. Heu, G.H., M.R.H., H.-Y. Huang, T.H., A.H., W.J.H., S.V.I., E.J., Z.J., C. Jones, S.J., C. Joshi, P.J., D.K., H.K., K.K., T. Khaire, T. Khattar, M. Khezri, M. Kieferová, S.K., A.K., P.K. A.N.K., F.K., J.M.K., D. Landhuis, B.W.L., P.L., K.-M.L., L.L.G., J. Ledford, J. Lee, K. W. Lee, Y.D.L., B.J.L., W.Y.L., A.T.L., W.L., W.P.L., A. Locharla, D. Lundahl, A. Lunt, S. Madhuk, A. Maloney, S. Mandrà, L.S.M., O.M., S. Martin, C.M., J.R.M., M.M., S. Meeks, K.C.M., A. Mieszala, S. Molina, S. Montazeri, A. Morvan, R.M., C.N., A. Nersisyan, M. Newman, A. Nguyen, M. Nguyen, C.-H.N., M.Y.N., W.D.O., K.O., A. Pizzuto, R.P., O.P., L.P.P., C.Q., M.J.R., D.M.R., G.R., C.R., E.R., N.C.R., N. Saei, K.S., K.J.S., H.F.S., C.S., M.J.S., A. Shorter, N. Shutty, V. Shvarts, V. Sivak, J. Skruzny, S. Small, W.C.S., S. Springer, G.S., J. Suchard, M.S., A. Sztein, D.T., A.T., M.M.T., A.V., S.V., B. Villalonga, C.V.H., S.W., S.X.W., T. White, K.W., B.W.K.W., C.X., Z.J.Y., P.Y., B.Y., J.Y., N.Y., G.Y., A.Z., N. Zhu, N. Zobrist, H.N., R.B., S.B., J.H., E.L., A. Megrant, J.K., Y.C., V. Smelyanskiy, G.V., P.R., A.M.L., D.A.A. and X.M. contributed to revising the paper and the Supplementary Information.

**Competing interests** A provisional patent application has been submitted for the analogue calibration scheme, titled 'High-accuracy calibration of an analog quantum simulator' (application number 63/639,509). The listed inventors are T.I.A., J.A.G., D.A.A. and X.M. The other authors declare no competing interests.

**Additional information**
**Correspondence and requests for materials** should be addressed to T. I. Andersen, D. A. Abanin or X. Mi.

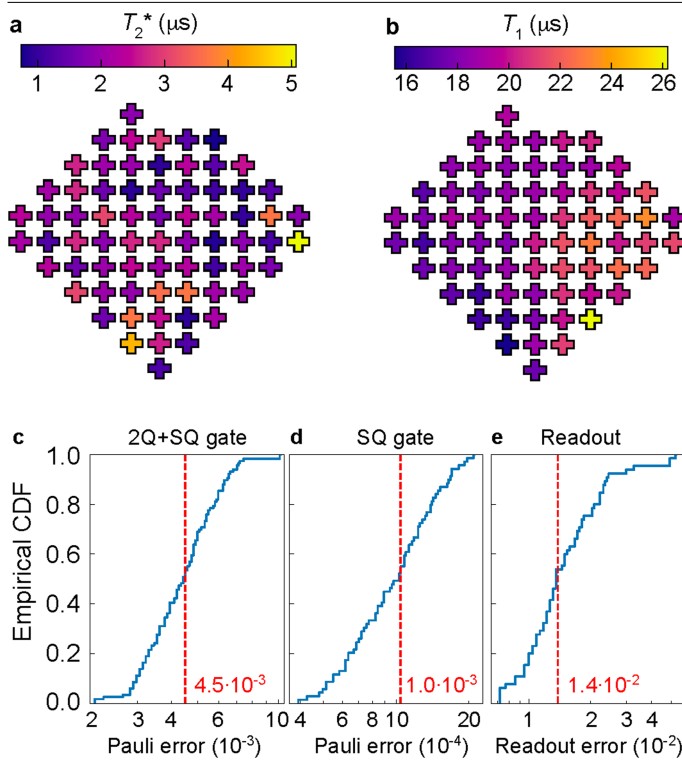

**a** $T_2^*$ (μs)

**b** $T_1$ (μs)

**c** 2Q+SQ gate  **d** SQ gate  **e** Readout

Empirical CDF

4.5·10⁻³ → $4.5 \cdot 10^{-3}$

1.0·10⁻³ → $1.0 \cdot 10^{-3}$

1.4·10⁻² → $1.4 \cdot 10^{-2}$

Pauli error ($10^{-3}$)  Pauli error ($10^{-4}$)  Readout error ($10^{-2}$)

**Extended Data Fig. 1 | Device characterization. a,b**, Ramsey dephasing ($T_2^*$; **a**) and photon relaxation ($T_1$; **b**) times across the qubit grid. **c,d**, Histogram of Pauli error for combined $\sqrt{\text{iSWAP}}$ and single qubit gates (**c**) and only single qubit gates (**d**). Red dashed lines indicate the median values. (CDF: cumulative distribution function). **e**, Histogram of readout errors.

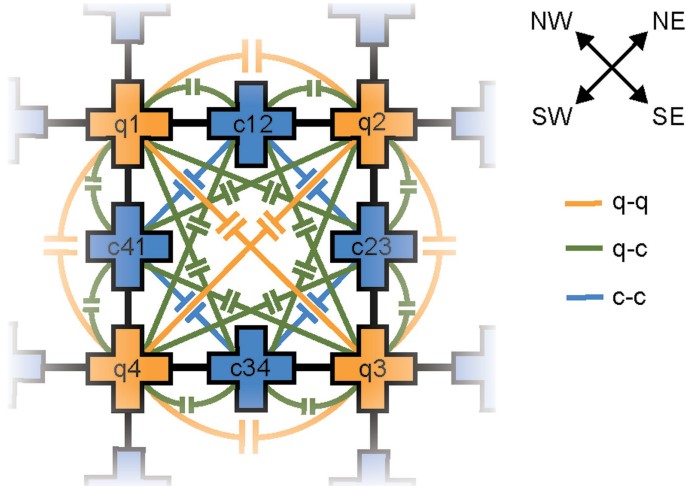

**Extended Data Fig. 2 | Schematic of underlying coupling pathways in the device.** In addition to capacitive coupling between neighboring qubits (orange) and couplers (blue), there are also diagonal next-nearest-neighbor couplings. Asymmetry in the underlying structure of the qubits causes a difference in the couplings along the NW-SE and NE-SW diagonals.

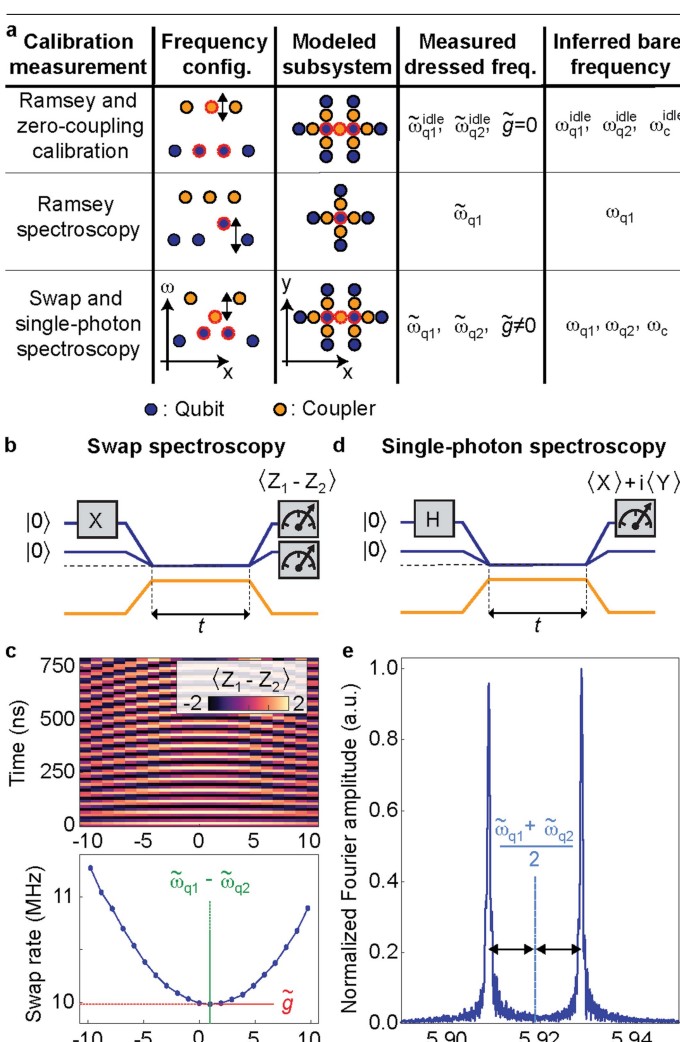

| Calibration measurement | Frequency config. | Modeled subsystem | Measured dressed freq. | Inferred bare frequency |
|---|---|---|---|---|
| Ramsey and zero-coupling calibration | | | $\widetilde{\omega}_{q1}^{idle},\ \widetilde{\omega}_{q2}^{idle},\ \widetilde{g}=0$ | $\omega_{q1}^{idle},\ \omega_{q2}^{idle},\ \omega_{c}^{idle}$ |
| Ramsey spectroscopy | | | $\widetilde{\omega}_{q1}$ | $\omega_{q1}$ |
| Swap and single-photon spectroscopy | | | $\widetilde{\omega}_{q1},\ \widetilde{\omega}_{q2},\ \widetilde{g}\neq 0$ | $\omega_{q1},\ \omega_{q2},\ \omega_{c}$ |

● : Qubit  ● : Coupler

**b  Swap spectroscopy**

$\langle Z_1 - Z_2 \rangle$

$|0\rangle$ — X

$|0\rangle$

$t$

**d  Single-photon spectroscopy**

$\langle X \rangle + i\langle Y \rangle$

$|0\rangle$ — H

$|0\rangle$

$t$

**c**

Top: $\langle Z_1 - Z_2 \rangle$, −2 to 2

Time (ns): 0, 250, 500, 750

Qubit detuning (MHz): −10, −5, 0, 5, 10

Swap rate (MHz): 10, 11

$\widetilde{\omega}_{q1} - \widetilde{\omega}_{q2}$

$\widetilde{g}$

**e**

Normalized Fourier amplitude (a.u.): 0.0, 0.2, 0.4, 0.6, 0.8, 1.0

$\dfrac{\widetilde{\omega}_{q1} + \widetilde{\omega}_{q2}}{2}$

Frequency (GHz): 5.90, 5.92, 5.94

**Extended Data Fig. 3 | Analogue calibration procedure. a**, Overview of calibration steps. We perform three main steps, which together allow for determining the bare frequencies of the qubits and couplers in the idle configuration in which $\widetilde{g}=0$ (top row), as well as in the interaction configuration (bottom two rows). For each step, we model a subsystem (third column) to convert the measured dressed frequencies (fourth column) to bare frequencies (fifth column). **b**, Circuit schematic of swap spectroscopy. **c**, Top: Measured population difference, $\langle Z_1 - Z_2 \rangle$, as a function of qubit detuning and time. Bottom: Extracted swap rate from Fourier transform vs qubit detuning. The position of the minimum allows for determining $\widetilde{g}$ and the difference of the dressed qubit frequencies, $\widetilde{\omega}_{q1} - \widetilde{\omega}_{q2}$. **d**, Circuit schematic of single-photon spectroscopy. **e**, Fourier transform of the measured $\langle X \rangle + i\langle Y \rangle$. The average of the peak positions is equal to the average of the dressed qubit frequencies $(\widetilde{\omega}_{q1} + \widetilde{\omega}_{q2})/2$.

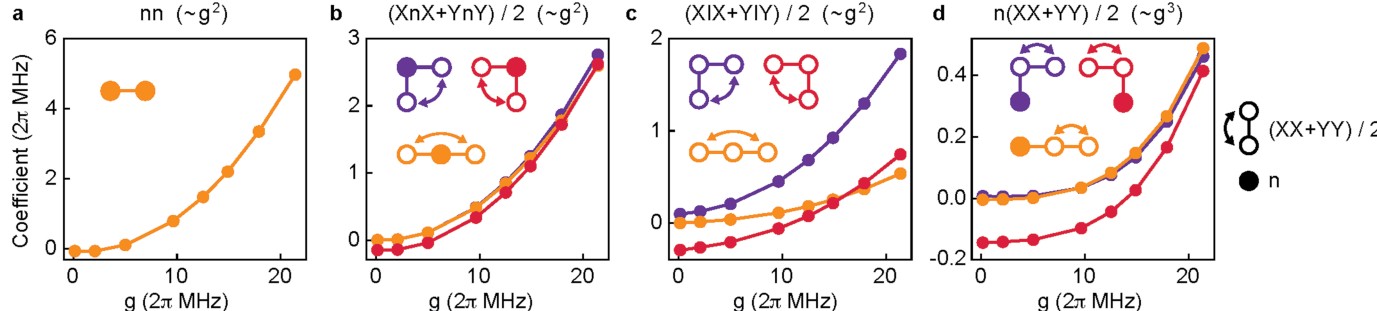

**Extended Data Fig. 4 | Higher order terms in the analogue spin Hamiltonian.**
Average coupling coefficient vs nearest-neighbor hopping $g$ for **a**, $n_i n_{i+1}$,
**b**, $(X_i n_{i+1} X_{i+2} + Y_i n_{i+1} Y_{i+2})/2$, **c**, $(X_i X_{i+2} + Y_i Y_{i+2})/2$, and **d**, $n_i(X_{i+1} X_{i+2} + Y_{i+1} Y_{i+2})/2$, where
qubits $i$, $i+1$, and $i+2$ are placed along a connected line. Aside from an offset
due to diagonal capacitive coupling, the first three terms scale as $g^2/\eta$, while the
fourth scales as $g^3\eta^2$, where $\eta$ is the anharmonicity. At $g = 2\pi \times 10$MHz, all
higher-order terms are smaller than $1 \times 2\pi$ MHz. In the three latter terms, there
is asymmetry between the three possible configurations displayed in the insets
(see text for details). Note that $n_i(X_{i+1} X_{i+2} + Y_{i+1} Y_{i+2})/2$ does not differ on average
from $(X_i X_{i+1} + Y_i Y_{i+1})n_{i+2}/2$ in **d**.

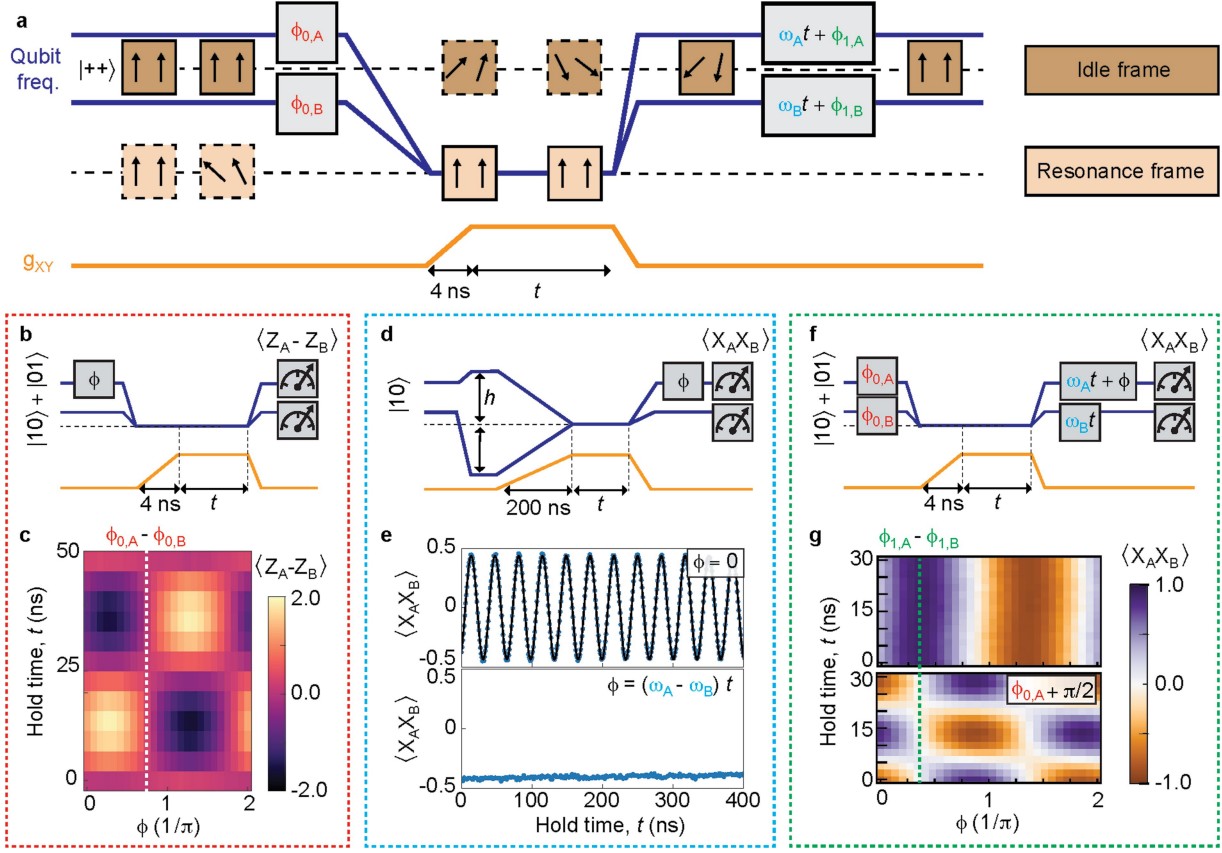

**Extended Data Fig. 5 | Phase calibration for hybrid analogue-digital experiments. a**, Schematic of phase accumulation and correction throughout hybrid analogue-digital circuit. While we typically prepare initial dimer states, we here consider an initial state $|++\rangle$ for the purpose of simplified explanation. Blue and yellow lines show qubit frequency trajectories and coupling profile, respectively, while brown (beige) boxes show the relative alignment of the two spins in the idle (resonance) frame. We apply corrective phases $\{\phi_{0,i}\}$ before the analogue circuit to ensure the correct dimer phase in the resonance frame when the analogue Hamiltonian is turned on. Additional phases $\{\omega_i t + \phi_{1,i}\}$ are applied after the analogue evolution in order to measure the same phase in the idle frame as was in the resonance frame. **b**, $\{\phi_{0,i}\}$ are calibrated by preparing triplet states, sweeping the phase difference within each qubit pair, and measuring the population difference after a variable time $t$. **c**, Population difference after time $t$ for an applied phase difference $\phi$. Since only the dimer phases 0 and $\pi$ are eigenstates of the analogue Hamiltonian, the correct $\phi_{0,i}$ is determined by minimizing the population oscillations. **d**, $\{\omega_i\}$ are calibrated by performing adiabatic ground state preparation with an initial staggered field and a slow ($25/g_m$) ramp, and measuring the $\langle XX \rangle$ correlations a time $t$ after the ramp. **e**, Top: $\langle XX \rangle$ after time $t$ when applying no corrective phase after the analogue evolution. Since the low-energy final state is known to have long-range correlations, the observed oscillations can be fit to extract the time-dependent part of the corrective phase after the analogue pulse. Bottom: $\langle XX \rangle$ after time $t$ when applying the corrective phase found from fitting the oscillations. The near-constant value indicates a successful correction. **f**, $\{\phi_{1,i}\}$ are calibrated by preparing an initial dimer state, performing the same circuit as in the experiment with corrective pre-analogue phases $\{\phi_{0,i}\}$ and partial post-analogue phases $\{\omega_i t\}$, applying a variable phase $\phi$ to one qubit in each pair, and measuring the $\langle XX \rangle$ correlations a time $t$ after the ramp. **g**, Top: $\langle XX \rangle$ after time $t$. Since the state is known to be the triplet state, the correct $\phi_1$ is found from maximizing $\langle XX \rangle$ correlations. Bottom: As a complementary technique, one can prepare the singlet state instead and find the $\phi$ that minimizes variations in $\langle XX \rangle$ correlations.

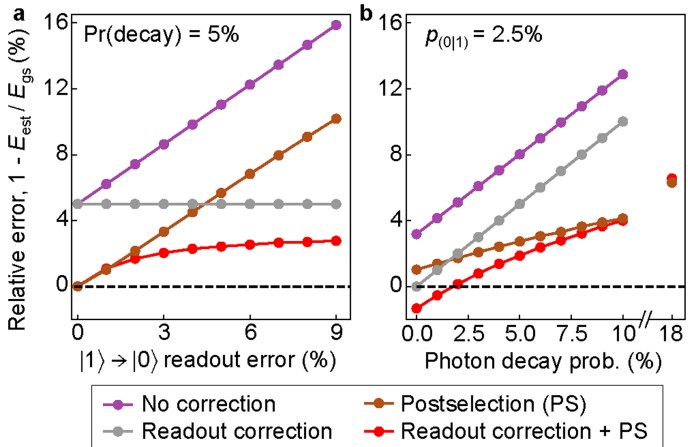

**Extended Data Fig. 6 | Correction for readout error and photon decay.**
Performance of various readout and photon decay correction techniques as a
function of **a**, readout error **b**, and photon decay probability. The performance
is measured as the relative error between the estimated energy ($E_{est}$) and the
actual ground state energy ($E_{gs}$). We find that the combined technique (red)
achieves the lowest relative error for a very wide range of parameters.

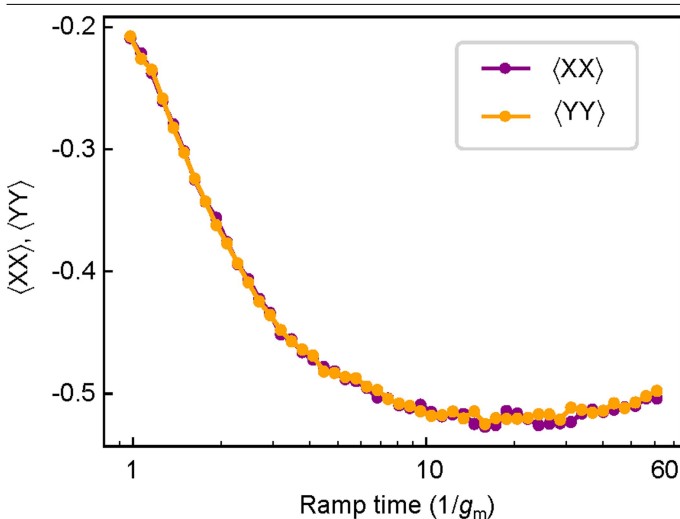

**Extended Data Fig. 7 | Comparison of *XX*- and *YY*-correlations.** Ramp time dependence of $\langle XX \rangle$ and $\langle YY \rangle$ averaged over all nearest-neighbor pairs. The two are found to be very similar, consistent with $U(1)$-symmetry.

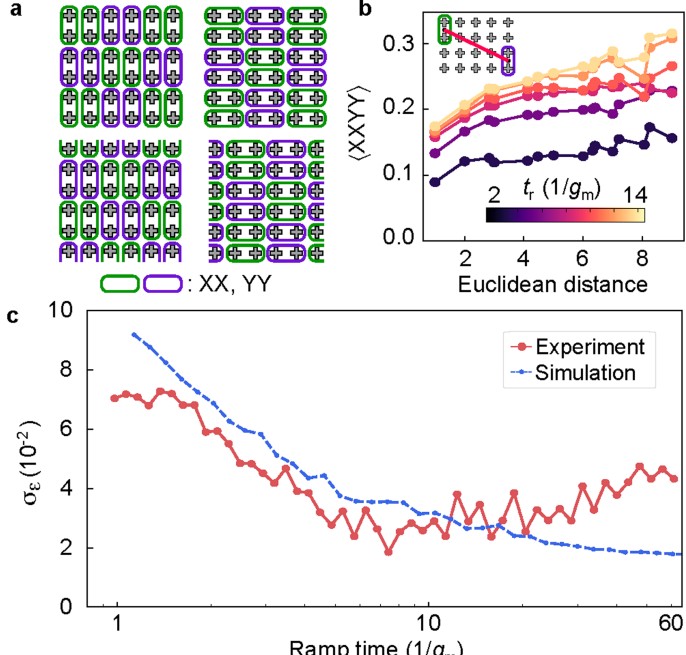

**a**

**b**

$\langle XXYY \rangle$

$t_r$ $(1/g_m)$

2 14

Euclidean distance

⬭ ⬭ : XX, YY

**c**

$\sigma_\varepsilon$ $(10^{-2})$

Ramp time $(1/g_m)$

— Experiment
— Simulation

**Extended Data Fig. 8 | Energy density fluctuations. a**, In addition to $\{X_i\}$, $\{Y_i\}$ and $\{Z_i\}$, we measure 8 periodic patterns of $XX$ and $YY$ to find $\sigma_\varepsilon$. **b**, $\langle XXYY \rangle$ has a relatively simple dependence on Euclidean distance (data from measurements shown in **a**), which can be interpolated to find the remaining terms. **c**, Energy density fluctuations, $\sigma_\varepsilon$, displaying good agreement between experiment (red) and simulation (blue); however, at long ramp times, decoherence causes higher fluctuations in the experimental case.

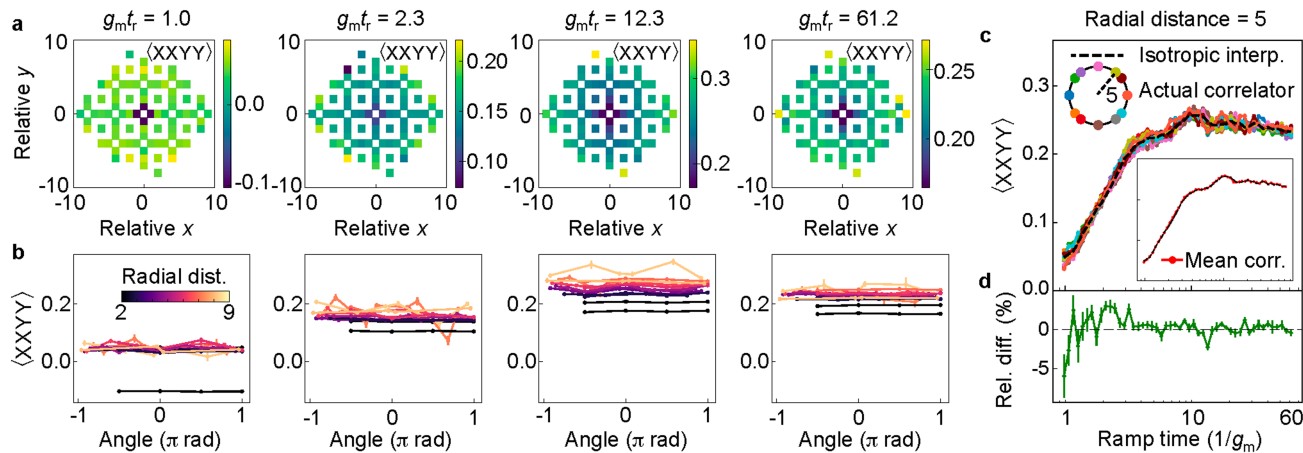

**Extended Data Fig. 9 | Isotropy of 4-qubit correlators. a**, Dependence of $\langle X_i X_j Y_m Y_n \rangle$ on relative position between centers of mass of sites $(i, j)$ and $(m, n)$, showing a substantial degree of isotropy. **b**, Angular dependence of $\langle X_i X_j Y_m Y_n \rangle$, displaying weak $\pi$-periodic oscillations that are most pronounced in the regime with longest correlations. **c**, Comparison of radial interpolation (dashed black) with the actual correlators at distance 5 (colored circles in main) and their average (dashed red in inset). **d**, Relative difference between radial interpolation and average of actual correlators, as a function of ramp time. The error is on the order of a few percent, and is comparable to the statistical error (error bars estimated from bootstrapping with 48,000 shots at each ramp time).

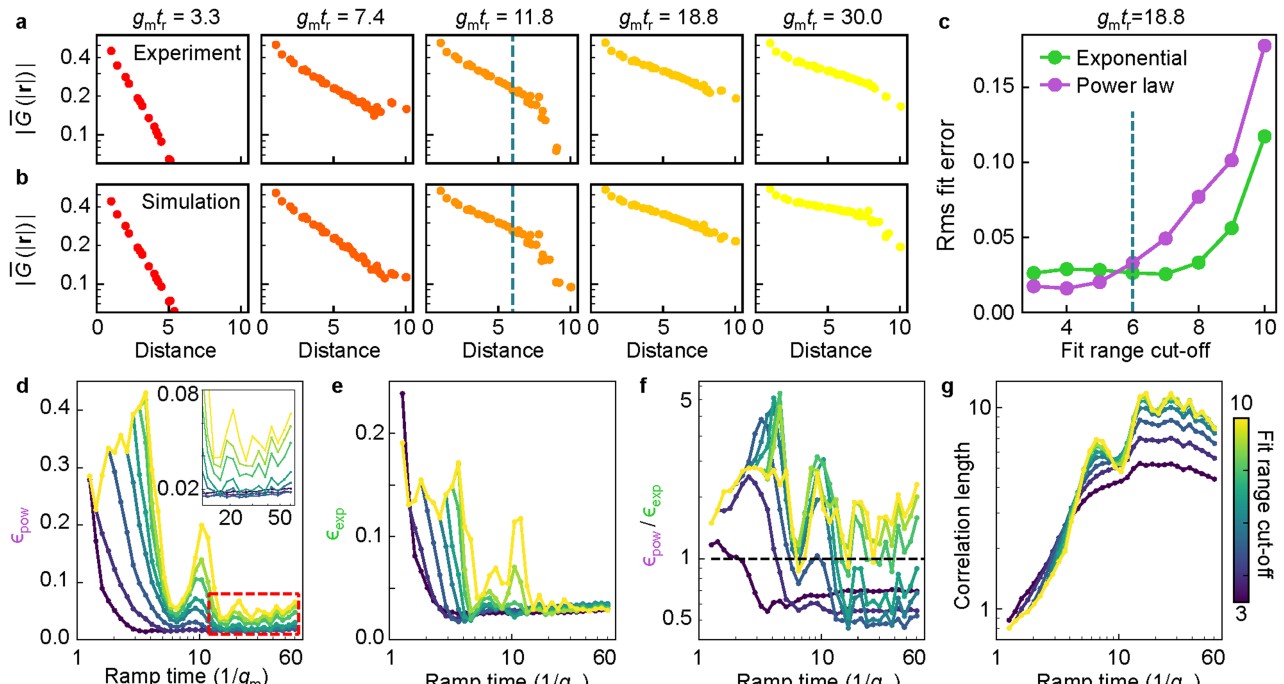

**Extended Data Fig. 10 | Correlation fitting. a,b,** Dependence of $\langle X_i X_j + Y_i Y_j \rangle / 2$ on Euclidean distance between $i$ and $j$ at various ramp times from experimental data (**a**) and simulation results (**b**). In both cases, we observe distortions at longer ($\gtrsim 6$ sites) distances, attributed to the finite size of our system. **c,** Root-mean-square fit error for exponential (green) and power-law (violet) fits, as a function of the cut-off distance applied in the fits. Going beyond a distance of 6, we observe a steep increase in the fit error, arising from the effects seen in **a**

and **b. d,e,** Ramp time dependence of rms error in power-law fits (**d**) and exponential fits (**e**) for various fit range cut-offs. Inset of **d**: Enlarged version of area indicated by red dashed square, showing abrupt increase in error for fit-range cut-offs longer than 7 sites. **f,g,** Ramp time dependence of rms error ratio (**f**) and correlation length (**g**). The drop in the fit error ratio below 1 is observed for all fit range cut-offs shorter or equal to 7 sites (**f**), and the discrepancy from KZ-scaling (dashed black) persists for all fit range cut-offs (**g**).