## [Peer Review File · Nature]

Thermalization and Criticality on an Analog-Digital Quantum Simulator

Corresponding Author: Dr Xiao Mi

Version 0:

Reviewer comments:

Referee #1

(Remarks to the Author)

The major technical development is a scalable protocol that makes possible calibration of the bare frequencies. The calibration error is estimated as 0.1 per qubit per evolution time of $1/g$. With a projection technique the Hamiltonian of the analog device can be approximated by the XY model to 10-20% accuracy. This effective model, plus post-selection techniques necessary to correct for photon losses, allow to study the Kibble-Zurek ramp dynamics as well as a variety of thermalization phenomena. For longer evolution times dephasing errors become appreciable that cannot be mitigated. In some instances heating due to the dephasing contributes to the final temperature that happens to be close to the estimated Kosterlitz-Thouless critical temperature. Two power laws are obtained that are consistent with the universality class of the KT transition.

Both the technical progress and the results on the K-Z mechanism and thermalization probably warrant publication in Nature after the following issue is addressed.

In case of the KZ ramps the dephasing is negligible up to ramp time $12/g$ when the final correlation length is estimated by an exponential fit to reach the size of 8 that is matching the system size. Not surprisingly there are finite size effects anticipated in parallel classical simulations. The KZ scaling of the correlation length with the ramp time t_r is claimed to be violated. This is not quite surprising after the full thermalization when the correlation length imprinted on the state during the transition can be washed away and all that remains is the excitation energy that depends on the ramp time. However, the observation that the correlation length continues to grow after the so-called freeze-out at the time $-\hat{t}$ before the critical point is not inconsistent with the KZ mechanism at all, see arXiv:1912.02815. I think that Fig. 3i should be replotted in a scaled form, i.e. as a scaled correlation length $|\xi|/\hat{\xi}$ in function of a scaled time t/\hat{t} , where $\hat{\xi}$ and $\hat{t} \propto \hat{\xi}^z$ are the K-Z length and time scales, respectively. Then, according to the KZ scaling hypothesis, the scaled plots for different ramp times t_r are expected to collapse in the KZ regime between $-\hat{t}$ and $+\hat{t}$. The collapse would demonstrate the KZ hypothesis. The supposed freezing of the correlation length in this regime is an artifact of the most naïve cartoon version of the KZ mechanism. In fact the length continues to grow and can increase a few times between $-\hat{t}$ and $+\hat{t}$. I expect that after the rescaling the plots in Fig. 3i will collapse almost up to their maxima. It is only what happens after the collapsed (KZ) regime that can be considered a post-KZ thermalization.

Referee #2

(Remarks to the Author)

In their manuscript entitled "Thermalization and Criticality on an Analog-Digital Quantum Simulator", Google Quantum AI and collaborators present an extensive analog quantum simulation study of a two-dimensional XY-type spin system using 69 superconducting qubits. In addition to performing native real-time evolution, in the form of quasi-adiabatic state preparation and quench dynamics, they leverage the universal digital capabilities of their device for measurements and state preparation. The main physical target of this study is thermalization of an XY magnet. In particular, the authors observe fast relaxation to semi-classical states consistent with a relatively high temperature; the build-up of critical correlations compatible with the expected Berezinskii-Kosterlitz-Thouless (BKT) phase transition, as well as deviations from predicted Kibble-Zurek (KZ) scaling behaviour which is attributed to interplay of quantum and classical coarsening; the cross-over from area- to volume- law entanglement after thermalization of entangled states with varying initial energy density, probing

aspects of the eigenstate thermalization hypothesis (ETH); and the transport of conserved quantities such as energy, in line with the emergence of hydrodynamics. In addition, they perform cross-entropy benchmarking (XEB), suggesting that the analog evolution lies beyond the reach of classical simulations.

This is an impressive work demonstrating not only the experimental capabilities of Google's Sycamore chip but also its precise calibration and characterization. Altogether, the present manuscript documents one of the most comprehensive quantum simulation experiments to date with full microscopic understanding. Many individual ingredients, however, are well known and have been presented elsewhere before. In more detail:

- i) The idea of hybrid digital-analog devices is not new (and has been reviewed, e.g. in Daley et al. 2022 [Ref. 9 of this manuscript]), but it is very much in line with the consensus of the quantum simulation community that high-fidelity analog evolution is (at least for now) typically superior to digital evolution.
- ii) Thermalization dynamics has been observed previously in quantum simulators such as the cited Refs. 18-25, where the (platform-dependent and non-universal) programmability of analog devices is often rather cleverly exploited to prepare resp. measure non-trivial initial states resp. observables (for example using multiple copies as by Kaufman et al. 2019 [Ref. 20]). Especially the use of randomized measurements is by now well established, especially in trapped-ion simulators. On the other hand, the use of precisely selected entangled initial states, as done here in the form of dimers, followed by analog dynamics, is rare.
- iii) Universal / critical aspects of real-time evolution – including standard KZ scaling, (classical) coarsening as well as less conventional self-similar dynamics – also have a long history in more traditional experiments using ultracold atoms, which is not reflected in the manuscript. For example: Sadler et al. Nature 443, 312 (2006); Braun et al., PNAS, 112, 3641 (2015); Navon et al. Science 347, 167 (2015); Prüfer et al. Nature 563, 217 (2018); Goo et al. Physical Review Letters, 128, 135701 (2022); .. just to name a few.
- iv) Related to iii), the emergence of conventional and unconventional hydrodynamic behaviour has also been probed in a variety of different experimental platforms, e.g. Sommer et al. Nature 472, 201 (2011); Zu et al. Science 376, 716 (2022); Joshi et al. Science 376, 720 (2022); ..
- v) Direct comparisons of extensive state-of-the-art simulations, indicative of analog quantum dynamics beyond the reach of classical computers have already been presented in the context of spin models, even for larger systems sizes than presented here, e.g. by Scholl et al. Nature 595, 233 (2021).

In view of the (incomplete) list of works mentioned above, I conclude that the main novelty of the present work lies in the technical results, namely the excellent characterization detailed device modeling of the analog Hamiltonian implemented by the Sycamore chip, together with tools more commonly employed in the context of digital quantum computing. This combination makes the present work stand out and warrants its publication. On the physics side, the most interesting observation is behaviour that appears to be incompatible with the standard KZ mechanism.

The paper is well written and especially the extensive supplement provides ample details for understanding the results presented in the main text. The abstract and introduction adequately reflect the content of the manuscript and set the stage, and most conclusions drawn are very convincing.

Concerning the methodology, the statistical data analysis of the experimental data presented appears to be incomplete as error bars are not visible throughout the whole manuscript and are also not mentioned in any figure caption! This flaw has to be remedied before considering publication. Although the results presented appear consistent, they cannot be fully assessed at this point due to this lack of error bars.

Finally, I have some more detailed comments and questions for the authors to consider in a revision of the manuscript, which are listed below.

- 1) The authors claim that “experimental calibration is only capable of resolving ‘dressed’ frequencies” and that “multi-parameter learning protocols are difficult to scale up”. This statement seems to contradict the efficiency of Hamiltonian learning techniques as discussed in Ref. 9.
- 2) In Fig. 2b, do the bitstring distributions shown correspond to the latest time in the inset (which is supposed to be $\sim 6/g$ according to the main text)? If not, why not? In the main text, it is claimed that the self-XEB converges to 1, but from the inset it looks like it reaches a lower value. Please comment on this. To make this small shift visible it might be better to plot the inset in a log-scale.
- 3) In Fig. 3e, the authors plot the ratio of rms errors for exponential and power law fits. From the data shown in Fig. 3d I am not entirely convinced about the robustness of this analysis. Are statistical errors in $G(|r|)$ taken into account here? How does the plot in Fig. 3e depend on the fit range (which seems to be chosen by hand)? What is the quality of the individual fits (rather than the rms ratio)? Have the authors tried to fit the generically expected combined algebraic + exponential decay, i.e. $G(|r|) \sim |r|^{-\gamma} \exp(-|r|/\xi)$ rather than both individually?
- 4) In relation to 3), I am wondering how much one can trust the extracted correlation lengths ξ shown in Figs. 3h & i. Especially given the insufficient statistical error analysis, I find it hard to evaluate whether or not the observed growth is truly far beyond KZ scaling. If I understand the analysis correctly, these correlation lengths are extracted from a simple exponential fit, although the power law character is clearly increasing in time, which could lead to a systematic time-dependent shift of ξ . Can the authors rule out such behaviour?
- 5) My concerns from comments 3) & 4) above aside, I find Fig. 3i, together with the corresponding points in 3h rather convincing. In the language of the cited Ref. 5 (Samajdar & Huse), I am wondering which of the scenarios 1-5 is actually realized in the present experiment. That is, does the ramp truly stop within the classical critical region as suggested in

Fig.3a, within or without the ordered phase? I think that a more thorough analysis and discussion of the physics at play here, also taken into account the infinite-order nature of the BKT transition would be suitable if the authors want to strengthen their claim of beyond-KZ dynamics. In this context, if the dynamics is not universal, can the authors quantify how much of the observed deviation comes from the additional Hamiltonian terms present in the experiment beyond Eq. (1)?

6) Can the authors provide more details about the diffusion model employed in the Fig. 5h? Is the diffusion constant D compatible with a simple (classical) thermal analysis or does it include more complex correlations that are classically intractable? In my opinion, the quantitative extraction of transport coefficients and the validation hydrodynamic models for quantum many-body models out of equilibrium is an excellent application for quantum simulators. A more thorough analysis of this point could therefore substantially strengthen the physical content of the present manuscript.

7) In the supplement A1, it is not clear to me which oscillators are truncated to 4 or 5 levels. Also, from the description of k_{qc} , it sounds as if the the couplings connecting $(q4,c23)$ as well as $(q1,c34)$ (though shown in Fig. S2) are not present. Is that correct?

8) In Fig. S4, especially the data in red, there is a substantial offset, i.e. the coefficients do not vanish as $g \rightarrow 0$. I therefore find the suggested scaling $\sim g^2$ or g^3 misleading. Can the authors explain this point?

9) Concerning the readout correction and postselection discussed in supplement D, I am confused about the dependence of the combined technique, whose relative error shown in Fig. S6 does not go to zero for decreasing readout error and photon decay probability. Why is that? If the mechanism is not properly understood, it seems that this correction protocol "accidentally" provides the best results in the range relevant to the experiment.

10) In supplement F, isotropicity is assumed. Given the inhomogeneities and asymmetric boundary conditions, can the authors quantify how well this assumption is fulfilled?

11) In the caption of Fig. S12, it says that the solid lines are the experimental results and the dashed lines show the global fit. This is probably a typo and supposed to be the other way around?

Version 1:

Reviewer comments:

Referee #1

(Remarks to the Author)

I am happy that rescaling figure 3i according to the KZ theory leads to a convincing collapse that extends even beyond the non-adiabatic stage between ν_{KZ} . If KZ theory is understood as a scaling hypothesis [see e.g. Phys. Rev. B 93, 075134 (2016)], rather than the cartoon adiabatic-impulse picture, then the collapse somewhat undermines the claim that the KZ theory breaks down after the phase transition. The issue how much it breaks down is currently under debate and this manuscript is an important experimental contribution to the discussion.

At a more technical level, ν_{KZ} used in the rescaling is obtained with a fit with the best exponent equal to 0.9 instead of the expected $\nu=0.67$. This discrepancy may be blamed on finite size effects but it should not be hidden from the reader and must be included in the manuscript, at best in the figure caption. I would recommend publication of the manuscript after this amendment.

Referee #2

(Remarks to the Author)

I thank the authors for their extensive reply, clarifications and modifications in response to my criticism. In my opinion, they satisfactorily addressed the concerns raised by the other referee and myself. Also the revised manuscript reflects this appropriately.

At this point I have no further questions and can recommend the manuscript for publication.

RESPONSE TO REFEREES

REFEREE #1

The major technical development is a scalable protocol that makes possible calibration of the bare frequencies. The calibration error is estimated as 0.1 per qubit per evolution time of $1/g$. With a projection technique the Hamiltonian of the analog device can be approximated by the XY model to 10-20% accuracy. This effective model, plus post-selection techniques necessary to correct for photon losses, allow to study the Kibble-Zurek ramp dynamics as well as a variety of thermalization phenomena. For longer evolution times dephasing errors become appreciable that cannot be mitigated. In some instances heating due to the dephasing contributes to the final temperature that happens to be close to the estimated Kosterlitz-Thouless critical temperature. Two power-laws are obtained that are consistent with the universality class of the KT transition.

Both the technical progress and the results on the K-Z mechanism and thermalization probably warrant publication in Nature after the following issue is addressed.

We thank the Referee for reviewing our manuscript and are pleased to hear that the Referee found that our work merits publication with the proper addressing of the comment below. In fact, this comment allowed us to demonstrate universal coarsening behavior in a clearer way, which we hope will enhance the impact of our work even further. We are very grateful for this suggestion!

In case of the KZ ramps the dephasing is negligible up to ramp time $12/g$ when the final correlation length is estimated by an exponential fit to reach the size of 8 that is matching the system size. Not surprisingly there are finite size effects anticipated in parallel classical simulations. The KZ scaling of the correlation length with the ramp time t_r is claimed to be violated. This is not quite surprising after the full thermalization when the correlation length imprinted on the state during the transition can be washed away and all that remains is the excitation energy that depends on the ramp time. However, the observation that the correlation length continues to grow after the so-called freeze-out at the time $-\hat{t}$ before the critical point is not inconsistent with the KZ mechanism at all, see arXiv:1912.02815. I think that Fig. 3i should be replotted in a scaled form, i.e. as a scaled correlation length $\xi/\hat{\xi}$ in function of a scaled time t/\hat{t} , where $\hat{\xi}$ and $\hat{t} \propto \hat{\xi}^z$ are the K-Z length and time scales, respectively. Then, according to the KZ scaling hypothesis, the scaled plots for different ramp times t_r are expected to collapse in the KZ regime between $-\hat{t}$ and $+\hat{t}$. The collapse would demonstrate the KZ hypothesis. The supposed freezing of the correlation length in this regime is an artifact of the most naïve cartoon version of the KZ mechanism. In fact the length continues to grow and can increase a few times between $-\hat{t}$ and $+\hat{t}$. I expect that after the rescaling the plots in Fig. 3i will collapse almost up to their maxima. It is only what happens after the collapsed (KZ) regime that can be considered a post-KZ thermalization.

As encouraged by the Referee, we have now substantially extended our analysis of the Kibble-Zurek breakdown, as well as comparisons between our observations and the predictions of previous theoretical works, such as [Samajdar *et al.*, 2401.15144], [Biroli *et al.*, PRE (2010)], and [Sadhukhan *et al.*, arXiv:1912.02815].

As mentioned by the Referee, it has been predicted that the dynamics can be described via a ramp time-independent function, $f(x)$, through the following rescaling:

$$\xi / \xi_{\text{KZ}} = f(\tau / \tau_{\text{KZ}}),$$

where τ_{KZ} and ξ_{KZ} are the freezing time and corresponding correlation length, respectively (the times are here relative to the critical point, i.e. $\tau = t - t_c$ and $\tau_{\text{KZ}} = |t_{\text{KZ}} - t_c|$).

Notably, by rescaling the data in Fig. 3i in the revised manuscript, we indeed find that the curves collapse very well (Fig. R1a below), in excellent agreement with the predictions of the Referee and the references above. Importantly, the collapse extends well beyond the quantum critical regime, $-\tau_{\text{KZ}} < \tau < \tau_{\text{KZ}}$, pointing to dynamical universality. This is outside the range where even extended Kibble-Zurek mechanisms predict collapsing behavior (e.g. Sadhukhan *et al.*, arXiv: 1912.02815), and is indicative of coarsening dynamics. We again thank the Referee for motivating the additional analysis that led to these insights!

We emphasize that the rescaling involves only a single parameter, ξ_0 , extracted from fits of $\xi(\tau = \tau_{\text{KZ}}) = \xi_0 (\tau_{\text{KZ}} / t_r)^\beta$ (rather than “manually enforcing” that $\xi(\tau = \tau_{\text{KZ}}) / \xi_{\text{KZ}} = 1$ by using the experimentally observed $\xi(\tau = \tau_{\text{KZ}})$ for each individual curve), thus making the collapse even more convincing. Indeed, we find that $\xi(\tau = \tau_{\text{KZ}})$ scales as a power-law with τ_{KZ} (see inset of Fig. R1a below), with a somewhat higher exponent, $\beta = 0.9$, than $\nu = 0.67$ - this was also observed in MPS simulations and is likely due to finite-size effects. (Note that the expected exponent is here $-\nu$ rather than z since we normalize by the ramp time).

To further study the functional form of the function f , it is convenient to divide the Hamiltonian ramp into three sections (see Fig. R1b below), namely (1) the adiabatic regime $\tau \ll -\tau_{\text{KZ}}$, (2) the critical regime ($-\tau_{\text{KZ}} < \tau < \tau_{\text{KZ}}$), and (3) the non-critical regime ($\tau \gg \tau_{\text{KZ}}$).

- 1) In the adiabatic regime, the system is expected to remain near the ground state and thus exhibit a correlation length given by $\xi(\tau) = \xi_{\text{KZ}} |\tau / \tau_{\text{KZ}}|^{-\nu}$, or $f(x) = |x|^{-\nu}$. We observe behavior similar to this prediction, shown in Fig. R1b, with only small deviations that we attribute to the finite size of our system.
- 2) In the second (critical) regime, it is more difficult to predict a simple functional form of the scaling function f , but the growth of ξ is expected to be of order unity. In the experiment, we find that the collapsed curves exhibit $f(1) / f(-1) = 2.3 \pm 0.2$.
- 3) Finally, for $\tau > \tau_{\text{KZ}}$, the curves collapse very well all the way up to $\tau / \tau_{\text{KZ}} = 3$, showing power-law-like behavior with an exponent close to 1. Importantly, in this regime, the coexistence of gapped and gapless modes due to spontaneous breaking of U(1)

symmetry is likely to cause differences from theoretical results in the literature, which largely focus on a quantum Ising model.

In the revised manuscript, we have changed Fig. 3 to include the rescaling described above, and have also added a new paragraph describing the resulting collapsing behavior. In the final paragraph, we now also address the important point about the co-existence of gapped and ungapped modes in the ordered phase, which we hope will inspire future theoretical work. Since the coarsening dynamics in this case is still not fully theoretically understood, we have made sure to not make any claims about the exact scaling dependence there, including the effects of the classical critical regime on the coarsening. We have also added a new supplementary section G that presents the above comparison of our results and theoretical predictions.

Finally, we have also confirmed that the discrepancy from the Kibble-Zurek scaling is not due to the higher-terms in the Hamiltonian. In Fig. R1c below, we show MPS simulations for both the full Hamiltonian and the pure XY-model, and find very similar behavior.

Fig. R1: Collapse following rescaling and robustness to higher-order Hamiltonian terms: **a**, Correlation length rescaled by ξ_{KZ} (as defined above), plotted against the rescaled time for various ramp durations (colored dots), leading to a notable collapse. Upper left inset: Measured correlation length at the theoretically predicted freezing point, displaying power-law behavior, $\xi(\tau=\tau_{KZ})=\xi_0(\tau_{KZ}/t_r)^\beta$ with $\beta=0.9$. Lower right inset: Original experimental data. **b**, Same as **a**, but with two-sided logarithmic axes for $\tau < -\tau_{KZ}$ and $\tau > \tau_{KZ}$. In the first regime, the data shows similar behavior to the theoretically expected scaling, $f(x)=|x|^{-\nu}$ with $\nu=0.67$ (purple dashed line). In the second regime, we find an increase $f(1)/f(-1)=2.3$. In the third regime, we observe power-law-like behavior with a (heuristic) exponent near $\eta=1$. **c**, MPS simulation of the ramp time dependence of the correlation length for the device Hamiltonian (blue) and the pure XY-model (green), showing very similar behavior with each other and with the experimental data (red), as well as a clear deviation from Kibble-Zurek scaling (dashed black) in both cases.

#REFEREE 2

In their manuscript entitled “Thermalization and Criticality on an Analog-Digital Quantum Simulator”, Google Quantum AI and collaborators present an extensive analog quantum simulation study of a two-dimensional XY-type spin system using 69 superconducting qubits. In addition to performing native real-time evolution, in the form of quasi-adiabatic

state preparation and quench dynamics, they leverage the universal digital capabilities of their device for measurements and state preparation. The main physical target of this study is thermalization of an XY magnet. In particular, the authors observe fast relaxation to semi-classical states consistent with a relatively high temperature; the build-up of critical correlations compatible with the expected Berezinskii-Kosterlitz-Thouless (BKT) phase transition, as well as deviations from predicted Kibble-Zurek (KZ) scaling behaviour which is attributed to interplay of quantum and classical coarsening; the cross-over from area- to volume- law entanglement after thermalization of entangled states with varying initial energy density, probing aspects of the eigenstate thermalization hypothesis (ETH); and the transport of conserved quantities such as energy, in line with the emergence of hydrodynamics. In addition, they perform cross-entropy benchmarking (XEB), suggesting that the analog evolution lies beyond the reach of classical simulations.

This is an impressive work demonstrating not only the experimental capabilities of Google's Sycamore chip but also its precise calibration and characterization. Altogether, the present manuscript documents one of the most comprehensive quantum simulation experiments to date with full microscopic understanding.

We thank the Referee for the very positive description of our work and, in particular, for recognizing the precision and comprehensiveness of our study.

Many individual ingredients, however, are well known and have been presented elsewhere before. In more detail:

i) The idea of hybrid digital-analog devices is not new (and has been reviewed, e.g. in Daley et al. 2022 [Ref. 9 of this manuscript]), but it is very much in line with the consensus of the quantum simulation community that high-fidelity analog evolution is (at least for now) typically superior to digital evolution.

We are happy to hear that the Referee also finds hybrid digital-analog devices to be a promising path in quantum simulation. We have done our best to reference previous literature on hybrid operation, including Ref. 9 as pointed out by the reviewer, and have now also added Refs. 40 [experiment: Bluvstein *et al.* (2021)] and 41 [theory: Lamata *et al.* (2018)]. We of course fully agree that the idea of hybrid devices in itself is not new, but we believe that our experimental demonstration represents an important step in its development.

ii) Thermalization dynamics has been observed previously in quantum simulators such as the cited Refs. 18-25, where the (platform-dependent and non-universal) programmability of analog devices is often rather cleverly exploited to prepare resp. measure non-trivial initial states resp. observables (for example using multiple copies as by Kaufman et al. 2019 [Ref. 20]). Especially the use of randomized measurements is by now well established, especially in trapped-ion simulators. On the other hand, the use of precisely selected entangled initial states, as done here in the form of dimers, followed by analog dynamics, is rare.

We thank the Referee for emphasizing the novelty of the flexible dimer state preparation enabled by our hybrid operation. Indeed, we are very excited about the possibilities unlocked by this capability. We fully agree with the Referee that thermalization dynamics in itself is a broad field that has received great attention in several impressive studies, and have now also added Refs. 33 [Scholl *et al.*, *Nature* (2021)] and 51 [Schuckert *et al.*, arXiv:2310.19869 (2023)], which further emphasize this point.

iii) Universal / critical aspects of real-time evolution – including standard KZ scaling, (classical) coarsening as well as less conventional self-similar dynamics – also have a long history in more traditional experiments using ultracold atoms, which is not reflected in the manuscript. For example: Sadler *et al.* *Nature* 443, 312 (2006); Braun *et al.*, *PNAS*, 112, 3641 (2015); Navon *et al.* *Science* 347,167 (2015); Prüfer *et al.* *Nature* 563, 217 (2018); Goo *et al.* *Physical Review Letters*, 128, 135701 (2022); .. just to name a few.

iv) Related to iii), the emergence of conventional and unconventional hydrodynamic behaviour has also been probed in a variety of different experimental platforms, e.g. Sommer *et al.* *Nature* 472, 201 (2011); Zu *et al.* 597, 45 (2021); Wei *et al.* *Science* 376, 716 (2022); Joshi *et al.* *Science* 376, 720 (2022); ..

Combined response to iii and iv: We completely agree with the Referee that KZ scaling and coarsening have also been the topic of some very nice previous studies. As is also pointed out by the Referee below, we introduce numerous novel aspects that have typically been unavailable in previous studies, including:

1) Very extensive probing techniques that allow for observing the KZ breakdown along with signatures of the Kosterlitz-Thouless transition in a spin model, effects of the eigenstate thermalization hypothesis, entanglement scaling, energy fluctuations, and real-time thermalization dynamics, to mention a few.

2) A very high fidelity, which is also directly quantified in cross-entropy benchmarking.

3) Hybrid digital-analog operation, enabling several of the measurements listed in (1), and opening for a wide range of future studies.

4) Perhaps most importantly - and partially thanks to the Referee's point (5) below - we have now also performed further analysis of the data in Fig. 3i and observe universal (collapsing) behavior during the KZ breakdown. This agrees very nicely with the theoretical predictions of Samajdar *et al.* (Ref. 5) and adds another important novel aspect to our study. We thank the Referees for inspiring this addition to the paper.

Finally, we have also addressed the Referee's comment by adding references 21 and 25 to the works by Navon *et al.* and Prüfer *et al.* mentioned by the Referee above, as well as references 19 [Roychowdhury *et al.*, *PRB* (2010)] and 20 [Biroli *et al.*, *PRE* (2010)], which discuss previous theoretical insights about the KZ breakdown.

v) Direct comparisons of extensive state-of-the-art simulations, indicative of analog quantum dynamics beyond the reach of classical computers have already been presented in the context of spin models, even for larger systems sizes than presented here, e.g. by Scholl *et al.* *Nature* 595, 233 (2021).

We agree with the Referee that there have been impressive previous studies that compare complex dynamics with classical simulations and find that the latter fail. It should be pointed out that most of these works do not directly quantify fidelity, which is a requirement to exclude the possibility that the dynamics can be well captured by cleverly simplified algorithms. In our work, we explicitly evaluate the infidelity and find that it is at the forefront of what has been achieved in the past (e.g. approximately 3 times lower than in a recent state-of-the-art analog study [Shaw *et al.*, Nature (2024)]). Combining the fidelity with known insights about the limited spoofability of random circuit sampling experiments in the regime that we operate in, we can evaluate the classical run-time with a higher degree of confidence. To ensure that our manuscript properly represents the impactful work that has been done previously on classically complex dynamics, we have now also included the work by Scholl *et al.* as Ref. 33 in the revised manuscript.

In view of the (incomplete) list of works mentioned above, I conclude that the main novelty of the present work lies in the technical results, namely the excellent characterization detailed device modeling of the analog Hamiltonian implemented by the Sycamore chip, together with tools more commonly employed in the context of digital quantum computing. This combination makes the present work stand out and warrants its publication. On the physics side, the most interesting observation is behaviour that appears to be incompatible with the standard KZ mechanism.

We thank the Referee for their kind description of the novel aspects of our work and for evaluating them as warranting publication.

The paper is well written and especially the extensive supplement provides ample details for understanding the results presented in the main text. The abstract and introduction adequately reflect the content of the manuscript and set the stage, and most conclusions drawn are very convincing.

We are pleased to hear the Referee's compliments about the writing of our manuscript and that the Referee finds the conclusions to be convincing.

Concerning the methodology, the statistical data analysis of the experimental data presented appears to be incomplete as error bars are not visible throughout the whole manuscript and are also not mentioned in any figure caption! This flaw has to be remedied before considering publication. Although the results presented appear consistent, they cannot be fully assessed at this point due to this lack of error bars.

Below, in Fig. R2.1, we have attached all relevant figures with error bars (markers omitted for clarity where necessary), computed by the bootstrapping technique. Seeing as we used ~10k-50,000k repetitions in the experiments and typically combine measurements of many observables, the statistical error bars are substantially smaller than the markers in almost all plots. In the revised manuscript, we have included error bars wherever they exceed the marker size, and the sentence "statistical error bars are smaller than marker size" in the caption in other cases. We have also commented on the statistical analysis and number of shots. We thank the Referee for encouraging this revision!

Fig 2**Fig. 3****Fig. 4****Fig. 5****Fig. R2.1: Additional statistical analysis.** Statistical errors, represented by error bars, are estimated from bootstrapping.

Finally, I have some more detailed comments and questions for the authors to consider in a revision of the manuscript, which are listed below.

1) The authors claim that “experimental calibration is only capable of resolving ‘dressed’ frequencies” and that “multi-parameter learning protocols are difficult to scale up”. This statement seems to contradict the efficiency of Hamiltonian learning techniques as discussed in Ref. 9.

We thank the Referee for making us aware that this part was not sufficiently specific. For completeness, let us first clarify the point regarding the dressed frequencies: since the system always exhibits some form of hybridization between qubits and couplers (as well as between qubits), any calibration measurement will only resolve a dressed quantity. Hence, in a typical local (single or two-qubit) calibration measurement, where nearby qubits and couplers are “turned off”, one measures a different dressed frequency than the one realized in the full-scale experimental setting when all qubits and couplers are turned to their “on-frequencies”. The dressed quantities are contrasted by the bare frequencies that we extract in our modeling, which would only be observable experimentally if all neighboring transmons were physically removed. Importantly, the bare frequencies do not change from local calibration measurements to global full-scale experiments, and are therefore much more useful.

Regarding the point about scalability, most *experimental demonstrations* of Hamiltonian learning so far have attempted multi-parameter optimization schemes, with cost functions including XEB fidelity [Roushan *et al.*, Neill *et al.*] and difference in single-qubit observables [Hangleiter *et al.*]. In these schemes, the growing number of parameters with system size poses a scalability challenge, and we were referring to these specific experimental demonstrations in the original version when claiming that previous studies resorted to “*multi-parameter learning protocols that are difficult to scale up*”. However, the Referee is completely right that more efficient solutions have been proposed, and we agree that these certainly deserve mentioning.

The reason for developing our new calibration technique instead of using these previously proposed schemes is a bit more involved and was therefore left out for brevity in the original version: experimentally, the Hamiltonian cannot be turned on instantaneously, and the unknown unitaries applied during the ramp-up and ramp-down scramble the information collected in most calibration measurements. In conversations with key contributors to the field of Hamiltonian learning, we typically found that this is a substantial obstacle. Moreover, these alternative schemes can often be quite sensitive to state preparation and measurement errors.

Importantly, spectroscopy measurements are highly robust to these difficulties (since we only measure the frequency). However, in relatively large subsystems, the number of eigenfrequencies is much smaller than the number of unknown Hamiltonian parameters, thus making spectroscopy ineffective. In contrast, for a subsystem of only 2 qubits (as used in our case), the combination of swap and single-photon spectroscopy allows for extracting the necessary information, thus making our scheme work very well.

We thank the Referee for this important comment, which we have now addressed by rephrasing the relevant part of the main text to (including also new references to Hamiltonian learning proposals):

“[...] Given this difficulty, past experimental studies either suffered from large errors or resorted to multi-parameter optimization protocols that are difficult to scale up. Sophisticated Hamiltonian learning techniques [Bailey et al., PRL (2019); Evans et al. arXiv:1912.07636] can circumvent these issues, but still have potential vulnerabilities to Hamiltonian ramps, noise, and errors in state preparation and measurement (SPAM).

In this work, we present a scalable calibration protocol that achieves low error by explicitly calibrating the bare frequencies. As illustrated in Fig. 1b, the protocol begins with two-qubit calibration measurements (single-photon and swap spectroscopy, which is robust to ramps and SPAM errors) to determine [...]”

2) In Fig. 2b, do the bitstring distributions shown correspond to the latest time in the inset (which is supposed to be $\sim 6/g$ according to the main text)? If not, why not? In the main text, it is claimed that the self-XEB converges to 1, but from the inset it looks like it reaches a lower value. Please comment on this. To make this small shift visible it might be better to plot the inset in a log-scale.

The shown bitstring distributions were measured at $t=6.1/g$, which is slightly later than the maximum of the inset, so we have now changed it to the latest time ($5.5/g$) and added this information in the caption. We thank the Referee for motivating this edit. The self-XEB departs from 1 at later times due to decoherence, which drives the system towards a uniform distribution, $\Pr(p) = 1/D$, with self-XEB equal to zero. We emphasize that the effects of decoherence are small, since the analog dynamics allow for fast thermalization. We have in fact already done our best to amplify this behavior near self-XEB=1, by plotting the inset in log-scale. To make it more visible, we have also added an expanded version of this plot to the SI as Fig. S16a, where the departure from self-XEB is even more clear.

In the main text, we describe the effects of decoherence in the sentence *“The observed fast scrambling dynamics are due to the simultaneously activated couplers, and - compared to an equivalent digital circuit - allow for less shift towards the decohered distribution, $\Pr(p)=D^{-1}$.* To avoid confusion, we now also explicitly mention the implication for self-XEB: *“[...] with vanishing self-XEB.”*

3) In Fig. 3e, the authors plot the ratio of rms errors for exponential and power-law fits. From the data shown in Fig. 3d I am not entirely convinced about the robustness of this analysis. Are statistical errors in $G(|r|)$ taken into account here? How does the plot in Fig. 3e depend on the fit range (which seems to be chosen by hand)? What is the quality of the individual fits (rather than the rms ratio)? Have the authors tried to fit the generically expected combined algebraic + exponential decay, i.e. $G(|r|) \sim |r|^{-(\gamma)} * \exp(-|r|/\xi)$ rather than both individually?

Below, we answer the Referee’s questions in a point-by-point manner:

Are statistical errors in $G(|r|)$ taken into account here?

We have performed statistical error analysis of the data presented in Fig. 3e, shown in Fig. R2.1 above. Using bootstrapping, we propagate the error all the way from the raw data to the final error ratio, and find that the decrease of the ratio below 1 is well within error.

How does the plot in Fig. 3e depend on the fit range (which seems to be chosen by hand)? What is the quality of the individual fits (rather than the rms ratio)?

We thank the Referee for providing us with the opportunity to give further details regarding the fitting procedure. Importantly, as shown in Fig. R2.2 below and also in some of the curves in Fig. 3d in the manuscript, finite-size effects distort the correlation decay at longer distances, observed both in experiment (a) and simulation (b). Specifically, at long distances, we find that the correlations drop rapidly for some ramp times and start increasing at others. If fitting up to long distances, these effects have a strong impact on the analysis, as can be seen from the sharp upturn in the fit error as we exceed a fit range of ~ 6 sites in Fig. R2.2c below. Informed by these findings, and the fact that the maximum distance at which such finite-size effects are still minimal is 6 sites, we decided to use this as our cut-off. We have added the figure below in the revised supplementary materials, in the newly added Supplementary Section D.

Fig. R2.2: Finite-size effects in correlations at long distances. **a,b**, Dependence of $\langle X_i X_j + Y_i Y_j \rangle$ on Euclidean distance between i and j at various ramp times from experimental data (**a**) and simulation results (**b**). In both cases, we observe distortions at longer (≥ 6 sites) distances, attributed to the finite size of our system. **c**, Root-mean-square fit error for exponential (green) and power-law (violet) fits, as a function of the cut-off distance applied in the fits. Going beyond a distance of 6, we observe a steep increase in the fit error, arising from the effects seen in **a** and **b**.

We address the Referee’s comment further by plotting the rms fit errors for all ramp times and a wide range of fit range cut-offs in Figs. R2.3a,b (power-law and exponential fits, respectively). From these plots, it is again evident that the fits with distance cut-offs longer than ~ 6 sites have particularly high errors. (It is of course natural to see some increase in error with increasing fit range, but we are here referring to the distinct increase seen especially well in the inset of Fig. R2.3a). Plotting the error ratio in Fig. R2.3c, we find that all fits up to a fit range of 7 sites display the same drop below a value of 1 around $g_m t_r = 10$. We also observe that the discrepancy from

KZ-scaling is maintained for all fit ranges. We have also added the figure below as Fig. S7, and thank the Referee for inspiring these valuable additions.

Fig. R2.3: Dependence of data analysis on fit range. **a,b**, Ramp time dependence of rms error in power-law fits (**a**) and exponential fits (**b**) for various fit range cut-offs. Inset of **a**: Enlarged version of area indicated by red dashed square, showing abrupt increase in error for fit-range cut-offs longer than 7 sites. **c,d**, Ramp time dependence of rms error ratio (**c**) and correlation length (**d**). The drop in the fit error ratio below 1 is observed for all fit range cut-offs shorter than or equal to 7 sites (**c**), and the discrepancy from KZ-scaling (dashed black) persists for all fit range cut-offs (**d**).

Have the authors tried to fit the generically expected combined algebraic + exponential decay, i.e. $G(|r|) \sim |r|^{-(\gamma)} \cdot \exp(-|r|/\xi)$ rather than both individually?

We did perform this type of fitting as well (see Fig. R2.4a below), and the results indeed nicely support the insights from the individual fits. As seen in Fig. R2.4b, the extracted correlation length from the exponential factor exhibits an abrupt increase in the same region as where the other KT-signatures are observed, indicating that the power-law behavior takes over. Moreover, in the same regime, the power-law exponent is found to reach a value even closer to $\frac{1}{4}$ than in the individual fits (Fig. R2.4c). Finally, we compare the relative dominance of the two behaviors in Fig. R2.4d, by plotting their decay contributions over the fit range ($d_{\min}=1, d_{\max}=6$):

$$\text{Exponential contribution: } \frac{1 - \exp[(d_{\min} - d_{\max})/\xi]}{1 - \exp[(d_{\min} - d_{\max})/\xi] \cdot (d_{\max}/d_{\min})^{-\gamma}},$$

$$\text{Power-law contribution: } \frac{1 - (d_{\max}/d_{\min})^{-\gamma}}{1 - \exp[(d_{\min} - d_{\max})/\xi] \cdot (d_{\max}/d_{\min})^{-\gamma}}.$$

Consistent with the conclusions drawn from the individual fits, we find that the decay contribution from the power-law strongly overtakes that of the exponential in the same low-temperature regime as where the other signatures are observed.

We fully agree that both techniques have their advantages, and the combined fit certainly complements the individual fits very nicely. For $t_r \geq 10/g_m$, the correlation length (ξ) extracted from the combined fit becomes much larger than the sample size and therefore provides little

information about the degree of decay across the device, and mainly shows that the power-law component has started becoming more dominant in the fit (which is already displayed in Fig. 3e). Because of this, we have decided to use the individual fits in the main text, and have added the combined fits, as well as the discussion above, in supplementary section E. We thank the Referee for inspiring this valuable addition.

Fig. R2.4: Combined exponential and power-law fits. **a**, Distance-dependence of correlations for various ramp times, fit with combined exponential and power-law fits (teal dashed lines). **b,c**, Ramp time dependence of correlation length (**b**) and power-law exponent (**c**), extracted from combined (teal), exponential (green) and power-law (purple) fits. **d**, Decay contributions, as defined above, for power-law (purple) and exponential (green) fits.

4) In relation to 3), I am wondering how much one can trust the extracted correlation lengths ξ shown in Figs. 3h & i. Especially given the insufficient statistical error analysis, I find it hard to evaluate whether or not the observed growth is truly far beyond KZ scaling. If I understand the analysis correctly, these correlation lengths are extracted from a simple exponential fit, although the power-law character is clearly increasing in time, which could lead to a systematic time-dependent shift of ξ . Can the authors rule out such behaviour?

We completely agree with the Referee about the importance of confirming that the observed breakdown is not sensitive to the statistical errors, fit range etc. Indeed, the discrepancy from KZ-scaling is found to be highly robust both to fit range (Fig. R2.3d above) and statistical error (see error bars in Fig. R2.1). In the revised version, we have now also added more emphasis on the fact that we do not focus on the longest ramp times in our analysis of the Kibble-Zurek breakdown:

“Focusing on shorter ramp times where these additional effects are absent and where the correlations exhibit a more clear exponential decay, we observe strong deviation [...]”

We now also comment on this robustness in Supplementary Section D. We thank the reviewer for helping us strengthen our claims in this way!

5) My concerns from comments 3) & 4) above aside, I find Fig. 3i, together with the corresponding points in 3h rather convincing. In the language of the cited Ref. 5 (Samajdar & Huse), I am wondering which of the scenarios 1-5 is actually realized in the present experiment. That is, does the ramp truly stop within the classical critical region

as suggested in Fig.3a, within or without the ordered phase? I think that a more thorough analysis and discussion of the physics at play here, also taking into account the infinite-order nature of the BKT transition would be suitable if the authors want to strengthen their claim of beyond-KZ dynamics. In this context, if the dynamics is not universal, can the authors quantify how much of the observed deviation comes from the additional Hamiltonian terms present in the experiment beyond Eq. (1)?

In the modified version, we have now substantially expanded on the relation between our observations and the predictions of Samajdar *et al.*, as well as the newly added reference 20 (Biroli *et al.*, PRE (2010)). We here provide an account of this comparison, along with answers to the Referee's specific questions.

Both works listed above predict that the coarsening dynamics can be described via a ramp-time independent function, $f(x)$, through the following rescaling:

$$\xi / \xi_{kz} = f(\tau / \tau_{kz}),$$

where τ_{kz} and ξ_{kz} are the freezing time and corresponding correlation length, respectively (the times are here relative to the critical point, i.e. $\tau = t - t_c$ and $\tau_{kz} = |t_{kz} - t_c|$).

Notably, by rescaling the data in Fig. 3i in the revised manuscript, we indeed find that the curves collapse very well (Fig. R2.5a below), in excellent agreement with the predictions of the references above. Importantly, the collapse extends well beyond the quantum critical regime, $-\tau_{kz} < \tau < \tau_{kz}$, pointing to dynamical universality. This is outside the range where even extended Kibble-Zurek mechanisms predict collapsing behavior (e.g. Sadhukhan *et al.*, arXiv: 1912.02815), and is indicative of coarsening dynamics.

We emphasize that the rescaling involves only a single parameter, ξ_0 , extracted from fits of $\xi(\tau = \tau_{kz}) = \xi_0 (\tau_{kz} / t_r)^\beta$, rather than "manually enforcing" that $\xi(\tau = \tau_{kz}) / \xi_{kz} = 1$ by using the experimentally observed $\xi(\tau = \tau_{kz})$ for each individual curve, thus making the collapse even more convincing. Indeed, we find that $\xi(\tau = \tau_{kz})$ scales as a power-law with τ_{kz} (see inset of Fig. R2.5a below) with a somewhat higher exponent, $\beta = 0.9$, than $\nu = 0.67$ - this was also observed in MPS simulations and is likely due to finite-size effects.

Regarding the Referee's question about the different scenarios introduced by Samajdar *et al.*, we observe signatures of the KT-transition (and energy right below the predicted transition point) for ramp times approximately in the range $10/g_m < t_r < 50/g_m$. Hence, for these ramp values, we expect a path through phase space that crosses through or ends in the classical critical regime, while for short ramp times, the system is expected to stay above the classical transition. Importantly, when comparing the dynamics to the scenarios introduced by Samajdar *et al.*, an important difference must be considered: while Samajdar *et al.* consider an Ising model that is fully gapped in the ordered phase, the XY-model exhibits a more subtle coexistence of gapped longitudinal modes and gapless transverse (Goldstone) modes, which affects the assumptions made about the equilibrium correlation length. Specifically, Goldstone

modes formally lead to an infinite correlation length at temperatures below the KT transition. Due to finite equilibrium correlation lengths on the ordered side in the Ising model, scenarios 2-5 in Samajdar *et al.* require stopping and waiting at the final points, which is different from our experimental protocol that does not involve waiting before the measurement. The complete effects of the coexisting gapped and gapless modes on the coarsening dynamics are yet to be fully understood, and we hope our observations will spark theoretical efforts to shed light on this subject.

To further study the functional form of the function f , it is convenient to divide the Hamiltonian ramp into three sections (see Fig. R2.5b), namely (1) the adiabatic regime $\tau \ll -\tau_{KZ}$, (2) the critical regime $(-\tau_{KZ} < \tau < \tau_{KZ})$, and (3) the non-critical regime $(\tau \gg \tau_{KZ})$.

- 1) In the adiabatic regime, the system is expected to remain near the ground state and thus exhibit a correlation length given by $\xi(\tau) = \xi_{KZ} |\tau / \tau_{KZ}|^{-\nu}$, or $f(x) = |x|^{-\nu}$. We observe behavior similar to this prediction, shown in Fig. R2.5b, with only small deviations that we attribute to the finite size of our system.
- 2) In the second (critical) regime, it is less trivial to predict f with a simple functional form, but the growth of ξ is expected to be of order unity. In the experiment, we find that the collapsed curves exhibit $f(1) / f(-1) = 2.3 \pm 0.2$.
- 3) Finally, in the non-critical regime in the ordered phase, the curves collapse very well all the way up to $\tau / \tau_{KZ} = 3$, showing power-law-like behavior with an exponent close to 1. We remind that this is the regime where the coexistence of gapped and gapless modes may cause differences from theoretical predictions based on the Ising model.

Fig. R2.5: Collapse following rescaling and robustness to higher-order Hamiltonian terms: **a**, Correlation length rescaled by ξ_{KZ} (as defined above), plotted against the rescaled time for various ramp durations (colored dots), leading to a notable collapse. Upper left inset: Measured correlation length at the theoretically predicted freezing point, displaying power-law behavior, $\xi(\tau = \tau_{KZ}) = \xi_0 (\tau_{KZ} / t_r)^{-\beta}$ with $\beta = 0.9$. Lower right inset: Original experimental data. **b**, Same as **a**, but with two-sided logarithmic axes for $\tau < -\tau_{KZ}$ and $\tau > \tau_{KZ}$. In the first regime, the data shows similar behavior to the theoretically expected scaling, $f(x) = |x|^{-\nu}$ with $\nu = 0.67$ (purple dashed line). In the second regime, we find an increase $f(1) / f(-1) = 2.3$. In the third regime, we observe power-law-like behavior with a (heuristic) exponent near $\eta = 1$. **c**, MPS simulation of the ramp time dependence of the correlation length for the device Hamiltonian (blue) and the pure

XY-model (green), showing very similar behavior with each other and with the experimental data (red), as well as a clear deviation from Kibble-Zurek scaling (dashed black) in both cases.

In the revised manuscript, we have changed Fig. 3 to now include the rescaling described above, and have also added a new paragraph describing the resultant collapsing behavior. In the final paragraph, we now also address the important point about the co-existence of gapped and ungapped modes in the ordered phase, which we hope will inspire future theoretical work. Seeing as the coarsening dynamics in this case is still not fully theoretically understood, we have made sure to not make any strong claims about the exact scaling dependence there, including the effects of the classical critical regime on the coarsening. We have also added a new supplementary section G that presents the above comparison of our results and theoretical predictions.

Finally, regarding the Referee's last question, we have indeed confirmed that the discrepancy from the Kibble-Zurek scaling is not due to the higher-terms in the Hamiltonian. In Fig. R2.5c, we show MPS simulations for both the full Hamiltonian and the pure XY-model, and find very similar behavior. We have now added this comparison in the supplementary materials, and thank the Referee for motivating this important clarification!

6) Can the authors provide more details about the diffusion model employed in the Fig. 5h? Is the diffusion constant D compatible with a simple (classical) thermal analysis or does it include more complex correlations that are classically intractable? In my opinion, the quantitative extraction of transport coefficients and the validation hydrodynamic models for quantum many-body models out of equilibrium is an excellent application for quantum simulators. A more thorough analysis of this point could therefore substantially strengthen the physical content of the present manuscript.

In the regime studied in our work, the observed behavior indeed appears to be compatible with a simple diffusive model. We thank the Referee for pointing out that the details of the model were unclear, and we have now added a new Supplementary section J, where we describe the model in further detail, reproduced below:

"In Fig. 5h in the main text, we fit the observed energy transport with a diffusion model, which we describe in further detail here. We define the energy density at site (i,j) , $e_{i,j}(t)$, as the average of the energy $\langle XX+YY \rangle / 2$ on the bonds that include site (i,j) and model the transport using a simple discretized version of the diffusion equation:

$$de_{i,j}(t)/dt = D(e_{i+1,j} + e_{i-1,j} + e_{i,j+1} + e_{i,j-1} - 4e_{i,j}),$$

where the diffusion constant, D , is the only fit parameter."

We completely agree with the Referee that the modeling of transport coefficients and study of hydrodynamic models is a very interesting application of our quantum simulator. While our study considers transport in a relatively high temperature regime, where the correlation length is

sufficiently short to allow simple classical calculations to be accurate, we expect transport at somewhat lower temperature to exhibit a richer behavior and to potentially have a higher classical complexity.

7) In the supplement A1, it is not clear to me which oscillators are truncated to 4 or 5 levels. Also, from the description of k_{qc} , it sounds as if the the couplings connecting (q_4, c_{23}) as well as (q_1, c_{34}) (though shown in Fig. S2) are not present. Is that correct?

We thank the Referee for making us aware of this potential source of confusion. The choice between 4 and 5 levels does not depend on the nature of the oscillator, but rather the Hamiltonian term we are interested in. Specifically, in calculations concerning 1- and 2-photon terms, we include 4 and 5 levels, respectively. This is done to account for effects that do not obey the rotating wave approximation, which couple $|1\rangle$ to $|3\rangle$ and $|2\rangle$ to $|4\rangle$. We agree that this was not sufficiently clear, and have now added the explanation above to the revised supplement.

In the description of the coupling efficiencies, we presented the minimal set of transmon pairs for brevity (the remaining pairs are given by symmetry). We have now addressed the Referee's comment by revising the text to make this more clear:

Notably, the geometry of the transmons breaks the 90° rotational symmetry; specifically, the couplings differ along the northwest-southeast (NW-SE) and northeast-southwest (NE-SW) directions. To discuss the three types of coupling, we consider the 4 qubits on a plaquette shown in Fig. S2 and consider representative examples of pairs of transmons (the formulas for the remaining pairs are given by reflection symmetry about the NW-SE and NE-SW axes, e.g.

$$\tilde{k}_{q_1, c_{23}} = k_{q_1, c_{23}} + 2k_{q_1, q_2} k_{q_2, c_{23}} \text{ infers that } \tilde{k}_{q_1, c_{34}} = k_{q_1, c_{34}} + 2k_{q_1, q_4} k_{q_4, c_{34}}):$$

8) In Fig. S4, especially the data in red, there is a substantial offset, i.e. the coefficients do not vanish as $g \rightarrow 0$. I therefore find the suggested scaling $\sim g^2$ or g^3 misleading. Can the authors explain this point?

Indeed, the Referee is completely right that there is an offset, which arises from the diagonal capacitive coupling. This coupling pathway is strongest along the NW-SE direction shown by the red curves, and offsets the coupling that arises from nearest-neighbor capacitive coupling. (Note that both the diagonal and nearest-neighbor couplings often enter the Hamiltonian through a virtual exchange interaction, which is why the diagonal coupling can induce an offset in the $(XX+YY)_n$ term in Fig. S4d, despite $XX+YY$ acting on neighboring qubits). The described scalings are for the contributions arising from nearest-neighbor capacitive coupling (and the resultant virtual exchange interactions). To address this point, we have now added the following sentence in the revised supplement:

"We note that there is an offset to these scaling behaviors, which arises due to the diagonal capacitive coupling. This is particularly evident for terms involving qubits along the NW-SE diagonal, since the diagonal coupling is strongest in that direction."

We have also modified the caption to:

“Aside from an offset due to diagonal capacitive coupling, the first three terms scale as [...]”

9) Concerning the readout correction and postselection discussed in supplement D, I am confused about the dependence of the combined technique, whose relative error shown in Fig. S6 does not go to zero for decreasing readout error and photon decay probability. Why is that? If the mechanism is not properly understood, it seems that this correction protocol “accidentally” provides the best results in the range relevant to the experiment.

We are pleased that the Referee has brought up this very interesting point. In our experiment, readout errors cause us to measure a higher energy than the actual energy of the quantum state. In addition, our photon number conserving Hamiltonian allows for postselecting measurements without T1 loss. Therefore, in order to evaluate the energy as accurately as possible, we have combined Bell state readout enabled by our hybrid analog-digital scheme, readout characterization, and a somewhat advanced correction scheme to reduce the effects of readout and T1 errors. As also made clear in the supplement, we completely agree that even this new technique does not remove the errors perfectly. However, it does allow us to quote a more accurate estimate of the actual energy of the quantum state than if using simpler correction techniques, across a very wide parameter range that extends well beyond that relevant in our experiment. As can be seen from Fig. R2.6, the error of our combined technique continues to remain very low at even higher readout errors as well, and converges to a value that is only slightly higher than that of pure postselection at very high T1 errors. In other words, we would like to emphasize that the combined technique is not only effective in a very particular regime.

We have now updated Fig. S11 with a simulation of 64 qubits (see Fig. R2.6 below), where we have also included a more realistic asymmetric readout error, in which $|1\rangle$ is measured as $|0\rangle$ more frequently than the other way around. We have also extended the range of parameters to emphasize that the combined technique is effective well beyond the parameter regime of our experiment.

Fig. R2.6: Readout and T1 error correction. Performance of various readout and photon decay correction techniques as a function of readout error (a) and photon decay probability (b), including an experimentally realistic readout bias of $p_{(0|1)}=3.7p_{(1|0)}$. The performance is measured as the relative error between the estimated energy E_{est} and the actual ground state energy E_{gs} . We find that the combined technique (red) achieves the lowest relative error for a very wide range of parameters that extends beyond the relevant range of our experiment.

Importantly, we have ensured that the technique is not strongly biased towards low energies; in fact, it still *overestimates* the energy for all parameters relevant to our experiment, and only slightly underestimates the energy in the limit of very low T1 errors. In this special case, an interesting effect occurs, which we explain by first reminding of the protocol: in a measured N-qubit bitstring (converted to the pairwise Bell basis before measurement), we change the (two-qubit) bitstring of each individual pair with probabilities inferred from the characterized readout errors. We do this instead of simply multiplying the distribution of the two-qubit bitstrings by the inverse readout matrix, because this enables us to subsequently perform postselection w.r.t. conserved total photon number on the resultant N-qubit bitstrings. While the stochastic compensation of readout errors perfectly re-establishes the correct distribution of two-qubit bitstrings (by construction of the probabilities with which we change the two-qubit bitstrings), each final N-qubit bitstring has a non-zero probability of having the wrong total number of photons, even in the case of zero T1 error. Now, as we show in much greater detail below, the lowest-energy two-qubit state, $|10\rangle\text{-}|01\rangle$ (converted to $|10\rangle$ by Bell conversion) has a slightly higher chance of being postselected than other two-qubit states. The result of this is a slight underestimate of the energy, which we again remind is 1) very small ($\sim 1\%$) and 2) not relevant for the parameters in our experiment. Below, for completeness, we provide a detailed account of how this effect arises.

Let's define $p(s_i = s)$ and $p(\hat{s}_i = s)$ as the probability distributions of, respectively, actual bitstrings s_i on pair i , and bitstrings \hat{s}_i measured after readout error and readout correction ($s \in \{(0, 0), (0, 1), (1, 0), (1, 1)\}$). We also introduce $e(s)$ as the energy of bitstring s , and

$p(PS|\hat{s}_i = s)$ as the probability that an *observed* bitstring s on qubit pair i survives postselection, renormalized such that $\sum_s p(PS|\hat{s}_i = s) = 1$. Since the *observed* total energy, \hat{E} , is just the linear sum of energies in each pair, we can then write:

$$\hat{E} = \sum_i \sum_s p(\hat{s}_i = s) e(s) p(PS|\hat{s}_i = s)$$

Now, crucially, in the limit of no T1 errors, we have $p(s_i = s) = p(\hat{s}_i = s)$, by construction of our Bayesian readout correction scheme with known readout errors. Hence, only non-uniformities in $p(PS|\hat{s}_i = s)$ across s can drive the energy away from its *true* value, $E = \sum_i \sum_s p(s_i = s) e(s)$. Said differently, the measured energy can only be wrong in the low-T1 limit if particular two-qubit bitstrings are postselected more often than others.

We next define N as the (correct) total number of photons in the system, $n(s)$, as the number of photons in a two-qubit bitstring s , and $\hat{N}_{-i} = \sum_{j \neq i} n(\hat{s}_j)$ as the observed number of photons in all qubits *except* pair i . The postselection probability is then given by the sum over all possibilities that give N photons in the total system, i.e. in the pair and remaining system combined:

$$p(PS|\hat{s}_i = s) = \sum_{s'} p(s_i = s' | \hat{s}_i = s) \cdot Pr(\hat{N}_{-i} = N - n(s) | s_i = s'),$$

where $p(s_i = s' | \hat{s}_i = s)$ is the probability that the actual bitstring on pair i was s' given that we measured s after readout error and readout correction. Moreover, $Pr(\hat{N}_{-i} = N - n(s) | s_i = s')$ is the probability that the remaining qubits were measured to have a total of $N - n(s)$ photons, given that the bitstring of pair i was actually s' . For system sizes exceeding ~ 10 pairs in a low energy state, we find to a good approximation that (N_{-i} is the *actual* number of photons in the remaining system):

$$Pr(\hat{N}_{-i} = N - n(s) | s_i = s') = Pr(\hat{N}_{-i} = N_{-i} + n(s') - n(s)) \equiv Q(n(s) - n(s')).$$

Intuitively, the state of the pair does not impact substantially the probability of making errors in the remaining system. In other words, we now find that the postselection probability is just a weighted sum of Q :

$$p(PS|\hat{s}_i = s) = \sum_{s'} p(s_i = s' | \hat{s}_i = s) \cdot Q(n(s) - n(s')),$$

where $\sum_{s'} p(s_i = s' | \hat{s}_i = s) = 1$ trivially. Now, we point out four important facts:

1) $Q(n(s) - n(s'))$ is maximized at $n(s) - n(s') = 0$, since intuitively the most likely outcome is to measure the right number of photons, $\hat{N}_{-i} = N_{-i}$, in the remaining pairs. (The difference

between $Q(0)$ and $Q(\pm 1)$ vanishes with increasing system size, causing a smaller biasing effect in larger systems.)

2) $Q(x) \sim Q(-x)$, since the combined effect of readout errors and incorrect compensation of readout imperfections is symmetric (even if the readout errors themselves are not).

2) $p(s_i = s' | \hat{s}_i = s)$ is negligible for $|n(s) - n(s')| > 1$ as long as readout errors are relatively small, since we are very unlikely to make a double error.

4) Under the constraint that $n(s) = n(s')$, we find that $\text{argmax } p(s_i = s' | \hat{s}_i = s)$ is $s = (1, 0)$, since the *observed* two-qubit bitstring that is most likely to be correct (or, more specifically, to have the right number of photons) is also the a priori most likely one, $(1, 0)$.

Bringing these four observations together, we find that the highest weighted sum (highest postselection probability) is achieved by $(1, 0)$. Since this is the two-qubit bitstring with lowest energy (after Bell conversion), we observe a negative bias in the case without T1 errors. We emphasize again that this effect is very small, and that we instead *overestimate* the energy at more experimentally relevant T1 errors.

10) In supplement F, isotropy is assumed. Given the inhomogeneities and asymmetric boundary conditions, can the authors quantify how well this assumption is fulfilled?

We completely agree with the Referee that it is good to discuss the role of isotropy in more detail. Below, in Fig. R2.7a, we show the spatial dependence of the XXYY-correlations (where the relative position is between the “center-of-mass” of the two qubit pairs, as in the supplement), displaying near-isotropic distributions. We observe a weak angular dependence with a period of π (Fig. R2.7b), which becomes most pronounced when the correlation length is maximized (see e.g. $g_m t_r = 12.3$ below). The amplitude is approx. ± 0.01 (or $\sim 5\%$ of the signal itself) and is expected to be due to the device shape.

Importantly, the only purpose of the assumption of isotropy is to interpolate radially before summing up all the correlators in our calculation of the energy fluctuations. Since we add all the correlators, this small degree of isotropy has very little effect, as we show next. In Fig. R2.7c, we compare the result of radial interpolation (dashed black curve) to the actual correlators at a radial distance of 5 sites (colored circles in main) and their average (red dashed curve in inset), and find that the difference is very small. In particular, we quantify the relative difference between the radial interpolation and the averaged actual correlators in Fig. R2.7d, and find that it is on the order of a few percent, and even smaller at the long times that are most essential to our conclusions. These deviations are comparable to the statistical noise (as shown by the error bars) and do not contribute a dominant effect to the total energy fluctuations. In the revised manuscript, we have added Fig. R2.7 as Fig. S14, as well as a discussion similar to the one above. We thank the Referee for motivating us to add this additional analysis!

Fig R2.7: Analysis of isotropy of correlations. **a**, Dependence of $\langle X_i X_j Y_m Y_n \rangle$ on relative position between centers of mass of sites (i, j) and (m, n) , showing a substantial degree of isotropy. **b**, Angular dependence of $\langle X_i X_j Y_m Y_n \rangle$, displaying weak π -periodic oscillations that are most pronounced in the regime with longest correlations. **c**, Comparison of radial interpolation (dashed black) with the actual correlators at distance 5 (colored circles in main) and their average (dashed red in inset). **d**, Relative difference between radial interpolation and average of actual correlators, as a function of ramp time. The error is on the order of a few percent, and is comparable to the statistical error (error bars estimated from bootstrapping with 48,000 shots at each ramp time).

11) In the caption of Fig. S12, it says that the solid lines are the experimental results and the dashed lines show the global fit. This is probably a typo and supposed to be the other way around?

We thank the Referee for noticing this typo, which we have now fixed.

Finally, we would like to thank the Referee for a careful review of our manuscript and for providing numerous very helpful comments. We believe that addressing them has allowed us to strengthen the impact and clarity of our messages, which will certainly improve the experience of readers.

Point-by-point response

Referee #1 (Remarks to the Author):

I am happy that rescaling figure 3i according to the KZ theory leads to a convincing collapse that extends even beyond the non-adiabatic stage between τ_{KZ} . If KZ theory is understood as a scaling hypothesis [see e.g. Phys. Rev. B 93, 075134 (2016)], rather than the cartoon adiabatic-impulse picture, then the collapse somewhat undermines the claim that the KZ theory breaks down after the phase transition. The issue how much it breaks down is currently under debate and this manuscript is an important experimental contribution to the discussion.

At a more technical level, ξ_{KZ} used in the rescaling is obtained with a fit with the best exponent equal to 0.9 instead of the expected $\nu=0.67$. This discrepancy may be blamed on finite size effects but it should not be hidden from the reader and must be included in the manuscript, at best in the figure caption. I would recommend publication of the manuscript after this amendment.

We again thank the Referee for reviewing our manuscript and for the very helpful suggestion regarding rescaling. To address the Referee's last point, we have now added information about the fitted exponent in the caption of Fig. 3, namely:

ξ_{KZ} is found from fitting $\xi(\tau=\tau_{KZ})=\xi_0(\tau_{KZ}/t_r)^{-\beta}$ with $\beta=0.9(1)$ (difference from $\beta=\nu=0.67$ likely due to finite-size effects).

Referee #2 (Remarks to the Author):

I thank the authors for their extensive reply, clarifications and modifications in response to my criticism. In my opinion, they satisfactorily addressed the concerns raised by the other referee and myself. Also the revised manuscript reflects this appropriately.

At this point I have no further questions and can recommend the manuscript for publication.

We are pleased to hear that the Referee found that our reply and revisions addressed their concerns, and that the Referee recommends the manuscript for publication.